# Pneumococcal competence is a populational health sensor driving multilevel heterogeneity in response to antibiotics

Marc Prudhomme [1,2,3], Calum H. G. Johnston [1,2,3], Anne-Lise Soulet[1,2], Anne Boyeldieu[1,2], David De Lemos[1,2], Nathalie Campo [1,2] & Patrice Polard [1,2] ✉

Competence for natural transformation is a central driver of genetic diversity in bacteria. In the human pathogen *Streptococcus pneumoniae*, competence exhibits a populational character mediated by the stress-induced ComABCDE quorum-sensing (QS) system. Here, we explore how this cell-to-cell communication mechanism proceeds and the functional properties acquired by competent cells grown under lethal stress. We show that populational competence development depends on self-induced cells stochastically emerging in response to stresses, including antibiotics. Competence then propagates through the population from a low threshold density of self-induced cells, defining a biphasic Self-Induction and Propagation (SI&P) QS mechanism. We also reveal that a competent population displays either increased sensitivity or improved tolerance to lethal doses of antibiotics, dependent in the latter case on the competence-induced ComM division inhibitor. Remarkably, these surviving competent cells also display an altered transformation potential. Thus, the unveiled SI&P QS mechanism shapes pneumococcal competence as a health sensor of the clonal population, promoting a bet-hedging strategy that both responds to and drives cells towards heterogeneity.

In all kingdoms of life, collective behaviour provides novel properties to a population unattainable at the individual level. Effective communication between individuals is key to achieving these features. In bacteria, such communication systems are crucial for survival, enabling an adaptive response to the selective pressure exerted by the niche on the population[1,2]. A widespread mode of bacterial cell-to-cell communication is quorum-sensing (QS), which relies on small exported molecules called autoinducers (AI). Historically, QS systems were shown to coordinate a cell population once the AI reaches a threshold, inducing gene expression changes and allowing an all-at-once populational response to environmental signals[3]. However, emerging studies on diverse QS systems have highlighted heterogeneity in their mechanisms, promoting differential coordination of the cell population[4–6]. Unravelling what drives the variability in such QS

mechanisms is central to understanding the behaviour of heterogeneously coordinated individuals. Here we addressed these two interlinked questions for the QS system that drives competence for natural transformation in *Streptococcus pneumoniae* (the pneumococcus).

Natural transformation is a key horizontal gene transfer mechanism in prokaryotes, unique in being entirely directed by the recipient cell. It involves the uptake of exogenous DNA followed by its chromosomal integration by homologous recombination, promoting intra- and inter-species genetic exchange[7,8]. In bacteria, transformation occurs during competence, which is developed and regulated in a species-specific manner[7]. Pneumococcal competence is transient and is controlled by the ComABCDE QS system, the AI of which is the exported Competence Stimulating Peptide (CSP)[9]. Remarkably,

[1]Laboratoire de Microbiologie et Génétique Moléculaires (LMGM), UMR5100, Centre de Biologie Intégrative (CBI), Centre Nationale de la Recherche Scientifique (CNRS), Toulouse, France. [2]Université Paul Sabatier (Toulouse III), Toulouse, France. [3]These authors contributed equally: Marc Prudhomme, Calum H. G. Johnston. ✉e-mail: patrice.polard@univ-tlse3.fr

competence transcriptionally modulates up to 17% of the ~2000 genes of the pneumococcal genome[10–12]. Pneumococcal competence provides cells with properties other than transformation, including the ability to kill non-competent relatives, defined as fratricide, which can be mediated by a competence-induced cell-wall hydrolase, CbpD[13]. Competent cells protect themselves from CbpD via production of the immunity protein ComM[14], which also promotes a transient division delay[15]. In addition, other properties are linked to competence, with the links undefined molecularly but important for the lifestyle of this pathogen in its human host. These include biofilm formation[16,17], virulence[18–22], increased survival upon transient antibiotic exposure[23] and host transmission[24].

The pneumococcal competence regulation cycle is well characterised at the single cell level. It begins with exported CSP, expressed from *comC* as a pre-peptide and matured by ComAB during its export. CSP activates the histidine kinase ComD, which in turn phosphorylates its cognate response regulator ComE (Fig. 1A)[25–28]. ComE-P induces expression of the early competence regulon, which includes the *comABCDE* genes, generating a positive feedback loop[28]. Two early competence genes encode the alternative sigma factor $\sigma^X$[29], which activates the late competence regulon, including 16 genes required for transformation. Among these is *dprA*, encoding a conserved late competence protein key for transformation[30], which also promotes pneumococcal competence shut-off after ~30 min (Fig. 1A, B)[31,32].

Another important characteristic of pneumococcal competence is its induction in exponentially growing cells in response to multiple types of exogenous stress. This key feature was unveiled with planktonic cells grown in an acidic medium, which prevents spontaneous competence development. In these non-permissive conditions, addition of exogenous stress can promote competence development[33–37]. However, how competence is coordinated within a population, either under permissive conditions or in response to stress, remains to be defined. In both cases, competence develops over time, unlike upon addition of synthetic CSP where competence is artificially synchronised. The most recent studies on spontaneous competence development, conducted in different strains, reported either a synchronised[38] or propagative[39] mechanism of populational competence development.

In this study, by combining genetic and single-cell analyses, we revisit the mechanism of competence development. We reveal a general mechanism of competence development irrespective of strain identity and for both spontaneous and stress-mediated competence. In this mechanism, the ComABCDE QS system can stochastically induce competence in response to stress in individual cells of a clonal population, which may lead to propagation through the population. Pneumococcal competence is thus a populational health sensor. We also show that this QS system promotes heterogeneity within a clonal population, generally increasing tolerance to lethal doses of antibiotics and transformation potential. In all, our results support the view of pneumococcal competence as more than a short-lived programme promoting natural transformation, revealing it as a bet-hedging strategy functionally diversifying a clonal population to face multiple types of environmental changes.

## Results

### Spontaneous pneumococcal competence occurs by propagation
The mechanism by which competence develops within a population is still debated, with one study in capsulated pneumococci supporting the historical view of competence as a classical QS system (Supplementary Fig. 1) where CSP diffuses and the whole population induces competence in a synchronous manner once a threshold is reached[38]. In contrast, we showed that competence propagates exponentially through a population by cell-to-cell contact in unencapsulated pneumococci[39]. This implies the existence of a subpopulation of cells

initiating competence, suggesting a bimodal mode of development. We proposed that heterogeneous stress levels in a population stochastically create this subpopulation, which perceives enough stress to self-induce competence linked to CSP retention on the envelope (Fig. 1C). This model promotes an exponential wave of competence through the population. We identified two key parameters, a fixed competence induction time ($X_A$) associated to the nature of the permissive environment but irrespective of inoculum density and a competence development rate ($X_B$, speed of propagation) correlated to the inoculum density (Fig. 1D)[39]. One hypothesis to explain the discrepancy between these studies was the presence of the polysaccharide capsule. We show that competence develops via propagation, irrespective of the presence of the capsule (Supplementary Fig. 2 and Supplementary Information)[39,40]. We demonstrate that an accumulation of genetic and hardware factors prevented visualisation of the early stages of competence development in the previous study performed with the capsulated D39 strain (Supplementary Figs. 3–5 and Supplementary Information)[38]. To further investigate the propagative model of spontaneous competence, we observed competence development at the individual cell level, using a cell biology approach visualising fluorescent Cy3-DNA binding to transforming cells[41]. This revealed propagation of competence throughout the population (Fig. 1E, F, Supplementary Fig. 6), mirroring the theoretical propagation curves presented (Fig. 1D). We further confirmed propagation of competence using sensitive single cell transformation assays (Supplementary Fig. 7 and Supplementary Information). Notably, the sensitivity of these assays revealed transformants prior to propagation, which may represent an initial subpopulation of self-inducing cells (Supplementary Fig. 7).

### Competence develops via Self-Induction and Propagation (SI&P) in response to antibiotic stress, shaping it as a populational health sensor
To further explore the existence of a self-inducing subpopulation at the origin of competence propagation, we tracked competence at the populational level via P*ssbB*::*luc* and at the cell level by sensitive transformation assays in non-permissive medium (Fig. 2A). This revealed a time window where stochastic low-level self-induction was detected only at the cell level, but no populational propagation was observed (Fig. 2B, top panels). In these conditions, competence is thus bimodal. As a control, we repeated this experiment with CSP added at 40 min, confirming that the ComABCDE QS system was able to respond in these conditions (Fig. 2B, top panels). It was previously shown that antibiotics induce competence in these non-permissive conditions[33]. To explore competence propagation in these conditions, we conducted this experiment in the presence of sublethal concentrations of streptomycin. Competence propagation was observed after 150 min (Fig. 2B, bottom panels), within the time window where self-induced cells were previously detected without propagation. To confirm that propagation developed within the entire population, this experiment was repeated with CSP added at 150 min, revealing competence induction with a steeper $X_B$, showing that CSP addition artificially synchronised the population (Fig. 2B, bottom panels). We deduced that in non-permissive medium, a stochastically self-induced subpopulation can be detected, but is insufficient to propagate competence throughout the population. Sublethal streptomycin addition, by increasing cellular stress levels, increases the self-inducing subpopulation, which reaches a threshold promoting competence propagation (Fig. 2B, bottom panels). Similar observations were obtained using Mitomycin C (MMC)[33] (Supplementary Figs. 8, 9 and Supplementary information). We unveil competence as a sensitive mechanism allowing individual cells to self-induce (SI) in response to stress, but only propagate (P) the signal in the face of sufficient stress. We call this mechanism Self-Induction and Propagation (SI&P). We have revealed that a growing pneumococcal culture is bimodal, with a

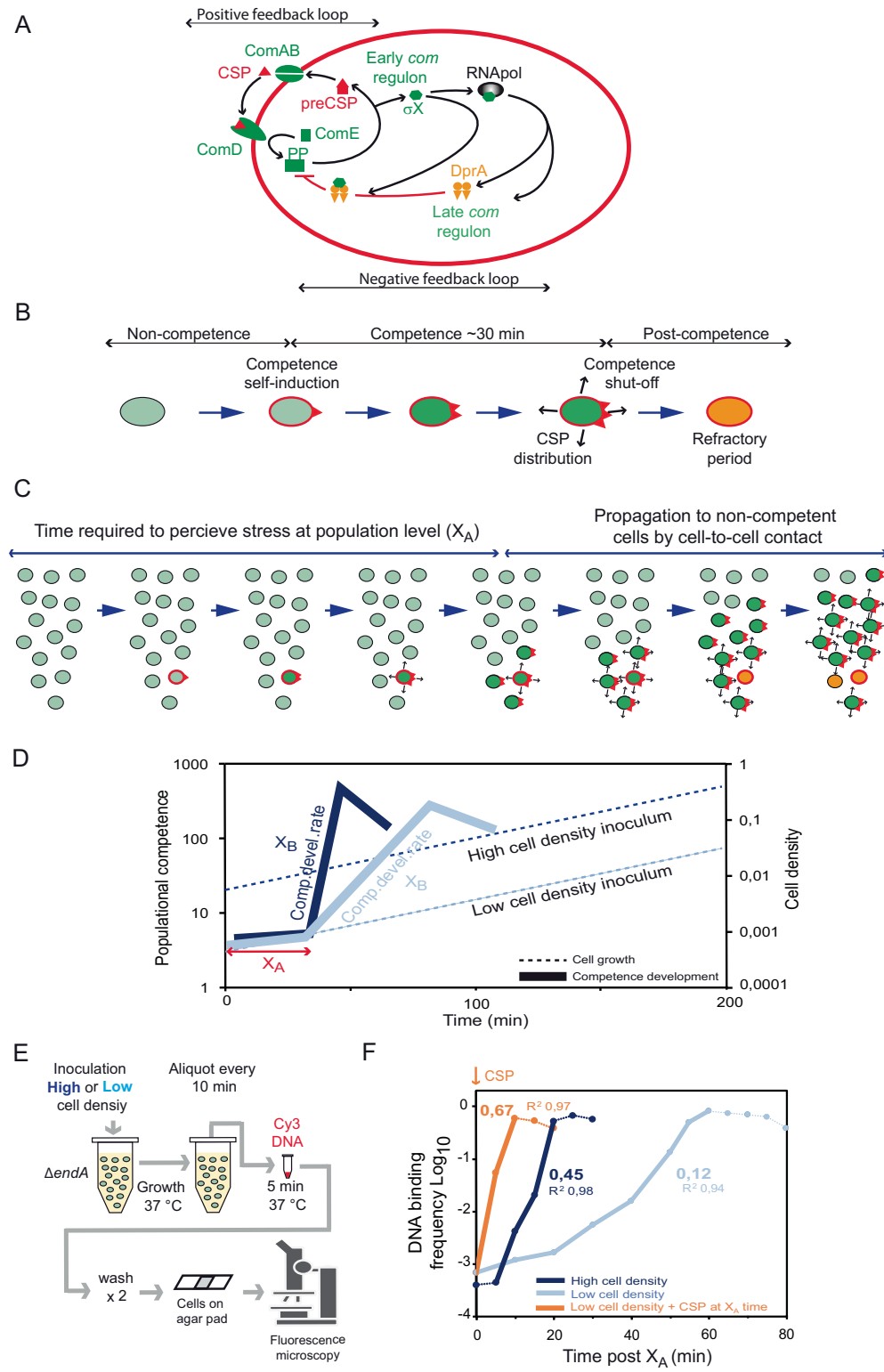

minority of cells self-inducing competence in response to stress (SI cells) and a majority of cells not reaching a threshold of stress high enough to stimulate self-induction and thus remaining non-competent, but able to be induced by propagation (P cells). This mechanism allows a pneumococcal population to gauge the threat of a stress, where the proportion of SI cells in a population dictates whether propagation of competence throughout a population occurs or not. The population can thus discriminate between genuine stress threat and false alarm. SI&P shapes pneumococcal competence as a population health sensor, able to evaluate the health of a population via environmental stress levels and react rapidly at the populational level only once a certain level of stress is reached.

**Fig. 1 | Pneumococcal competence development. A** Competence regulation at the individual cell level. Pre-CSP is matured and exported by ComAB and interacts with the histidine kinase ComD at the cell pole. ComD phosphorylates its cognate response regulator ComE and ComE-P dimers induce the early *com* regulon, including both *comX* genes, which encode the alternative sigma factor $\sigma^x$. $\sigma^x$ interacts with RNA polymerase to induce late *com* genes, including *dprA*. DprA is driven to the cell pole by $\sigma^x$ and interacts with ComE-P to promote competence shut-off. **B** Competence development at the individual cell level. A non-competent cell senses a stress (red contour) and becomes competent. This cell becomes a CSP donor, able to transmit competence to neighbouring cells by cell-to-cell contact. After ~30 min, the cell enters post-competence and is temporarily unable to respond to a competence signal. **C** Model of populational competence development by propagation. In a growing population, stress and metabolic heterogeneity creates a subpopulation of competent cells able to propagate competence to non-competent neighbours. These cells become competent and can in turn transmit competence, leading to exponential propagation throughout the population.

**D** Model of spontaneous propagation reported by the luciferase transcriptional fusion under the $P_{ssbB}$ late competence promoter ($P_{ssbB}::luc$)[40,80–83]. A time ($X_A$) is required to produce a cell fraction developing competence in an autocrine mode, which depends on environmental conditions and genotype. The competence development rate ($X_B$) is conditioned by the speed of propagation among non-competent cells, linked to cell density and CSP-retention ability. **E** Competence propagation visualised by microscopy. Observing binding of fluorescent tDNA to competent cells in different densities of inocula supports the propagation model. Non-competent cells were inoculated in permissive C + Y medium at $8 \times 10^{-3}$ (high cell density) or $8 \times 10^{-4}$ (low cell density) to allow visualisation of fluorescent Cy3-DNA bound by competent cells. **F** Frequency of cells binding Cy3-labelled tDNA throughout the culture from the $X_A$ time (30 min, Supplementary Fig. 5A). The rate of increase of tDNA-binding cells is reported with the $R^2$. Thick lines show data range used for rate calculations. Individual data shown representative of triplicate repeats.

## An artificially initiated competent cell fraction can propagate competence to non-competent cells

To explore the fraction of SI cells required to propagate competence, we attempted to identify the minimal artificial fraction of SI cells able to propagate competence to non-competent P cells. A wildtype non-competent cell culture was split in two and one sample was decorated with CSP by exposure during 1 min to mimic SI, and immediately washed at 4 °C to remove excess CSP not bound to the cells. The two samples were then mixed to generate varying ratios of artificial SI and P cells in permissive medium (Fig. 2C). The P cell population alone provided a sufficient fraction of SI cells to propagate competence via spontaneous induction after an $X_A$ time of ~70 min (Fig. 2D and Supplementary Fig. 9C). The addition of artificial SI cells should shorten the $X_A$ time, since the threshold of SI cells required for propagation will be reached more rapidly. The $X_A$ time was compared to a negative control where CSP was added to cell-free culture, washed and used in the same ratio (Fig. 2C) and a positive control of only artificial SI cells, which showed an $X_A$ time of <5 min, consistent with immediate competence induction through artificial synchronisation (Fig. 2D and Supplementary Fig. 9C). The mixed cultures showed a gradient of $X_A$ values, revealing a clear correlation between cell ratio and $X_A$ (Fig. 2D). As the ratio decreased, the $X_A$ increased until no clear effect could be attributed to the addition of artificial SI cells. A difference was observed for the two least diluted ratios ($4.8 \times 10^{-4}$, $4.8 \times 10^{-3}$), showing that the minimal SI cell fraction required to promote populational competence is found between the $4.8 \times 10^{-4}$ and $4.8 \times 10^{-5}$ cell ratios. In these experiments, competence development results from a critical ratio of artificial SI cells, mimicking the SI cell fraction. This fraction can promote propagation of the competence signal through a population, further supporting SI&P as the mode of regulation of the ComABCDE QS system.

## Competence improves tolerance to the genotoxic agents norfloxacin and methanemethylsulfonate

In our conditions, one SI cell in $10^{-5}$ to $10^{-6}$ could be enough to propagate competence to P cells (Fig. 2). We explored the benefits that artificially inducing competence in P cells could provide. A previous study reported that competence improved survival of pneumococci upon transient exposure to three competence-inducing antibiotics[23]. We further explored this competence-mediated benefit on cell survival by using two genotoxic drugs, norfloxacin (Norflo) and methanemethylsulfonate (MMS), which damage the replicating genome by blocking the gyrase or alkylating and depurinating nucleotides in DNA, respectively[42–44]. Both induce competence of exponentially growing cells at sublethal concentrations[33] (Supplementary Fig. 10A).

To evaluate how survival of competent P cells was impacted by these two stresses, we measured colony-forming units (cfu) following exposure at above minimal inhibitory concentrations (MIC) (Supplementary Fig. 10BC and Supplementary Table 1). We used strains which cannot naturally develop competence but can be induced by CSP, allowing rapid populational competence coordination (Methods, Supplementary Table 3). Cells activated (or not) during 10 min by CSP were exposed to above MIC concentrations of Norflo (100 µg mL$^{-1}$) during 30 min or MMS (625 µg mL$^{-1}$) during 15 min to compare survival (Fig. 3A and Supplementary Fig. 11A). Survival of non-competent cells reached 15% and 1%, respectively. By comparison, competent cells displayed improved survival, expressed as a tolerance ratio of 3.4 for Norflo and 4.3 for MMS (Fig. 3B). These experiments elaborated on the previous finding[23] that competence increases tolerance of cells faced with drugs applied transiently above their MIC.

Next, we explored how the timing of stress exposure relative to CSP influences this improved tolerance. We added Norflo or MMS either 5 min before CSP or concurrently with CSP and measured the cfu following the same incubation time with these two drugs as above (Supplementary Fig. 11B, C). Results showed that addition of CSP and stress concurrently reduced the tolerance ratio for Norflo and MMS (Fig. 3C). Stress addition prior to CSP further reduced the tolerance ratio for MMS (0.9) but not Norflo. These findings show that the timing of stress exposure relative to competence induction modulates the beneficial effect of competence differently depending on the stress. To investigate whether the improved survival afforded to a pneumococcal population extended beyond the ~30 min competence window[45], we removed excess CSP 10 min after addition to prevent further cycles of competence (as can be observed with several peaks of transformation with CSP addition at 40 min in Fig. 2B). Cells were then exposed to Norflo or MMS 30 or 50 min after CSP addition to assess survival of post-competent cells (Supplementary Fig. 11D, E). Results showed that post-competent cells maintain increased tolerance to MMS, but not Norflo (Fig. 3D). This further highlighted the heterogeneous effect of competence on descendent cell survival. This phenomenon can be visualised by plotting the tolerance ratio against the time of stress addition relative to CSP (Fig. 3E).

Exposure to MMS for 15 min was sufficient to kill ~99% of the population (Fig. 3B), defining the minimal duration for killing 99% of a population (MDK$_{99}$), a value used as a metric of bacterial tolerance[46,47]. In contrast, the MDK$_{99}$ was not established for Norflo as survival remained around 15% (Fig. 3B). To establish the MDK$_{99}$ for Norflo, a time course was carried out with survival of non-competent and competent cells compared at varying time-points after Norflo addition. MDK$_{99}$ values of ~74 and ~103 min exposure were respectively determined (Supplementary Fig. 10D). In conclusion, competent cells display improved tolerance to the genotoxic stresses MMS and Norflo. In addition, tolerance ratios are modulated dependent on when stress addition occurs compared to competence induction, highlighting a

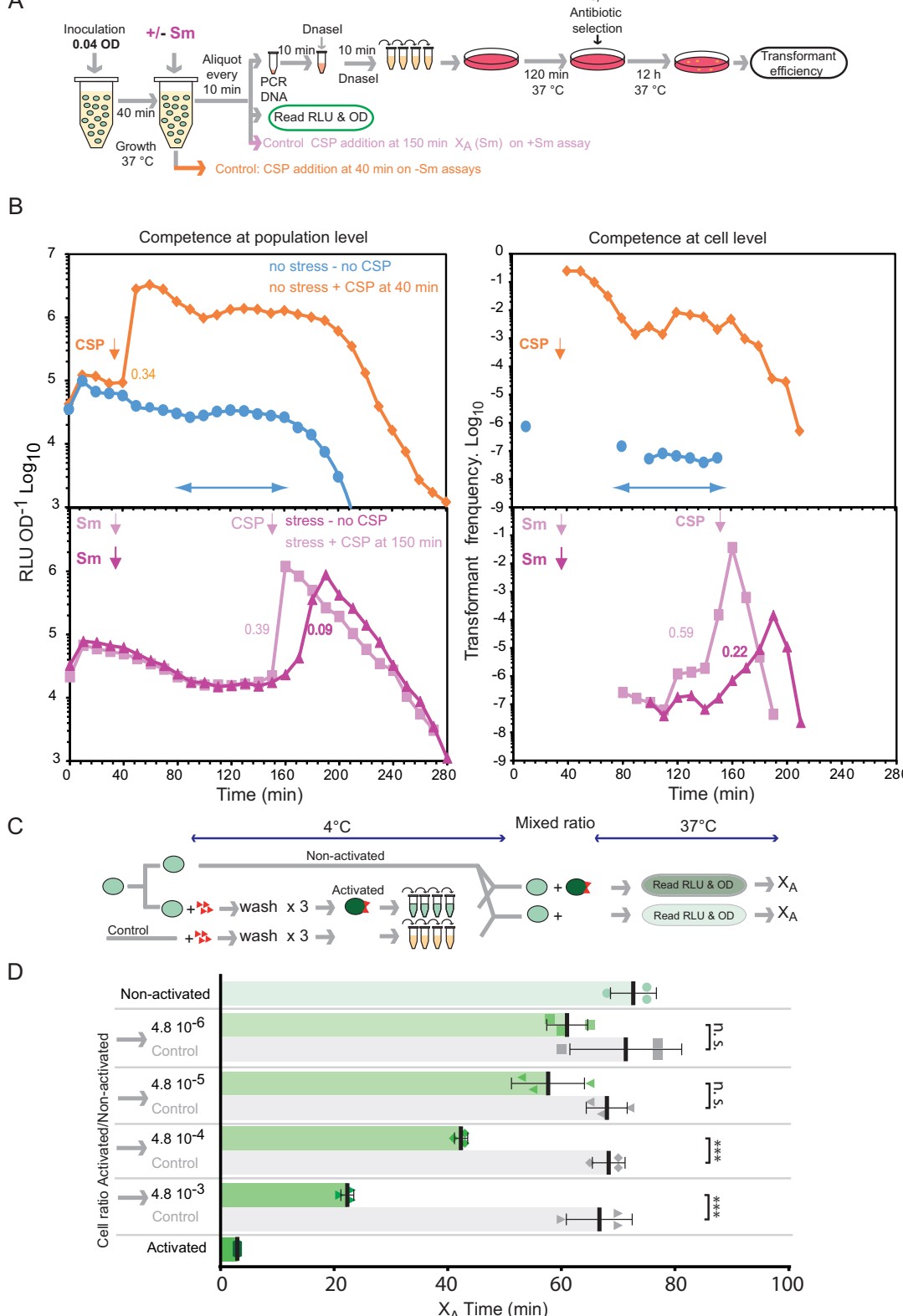

heterogeneous behaviour of competent cells in overcoming these stresses.

## Competence modulates survival of cells exposed to a wide variety of lethal stresses

To further investigate tolerance of competent cells faced with lethal stresses, we enlarged the analysis to different antibiotics, targeting basic cellular functions: cell wall synthesis (ampicillin, vancomycin), genome integrity (mitomycin C, novobiocin, trimethoprim), transcription (rifampicin) and translation (erythromycin, tetracycline, streptomycin, kanamycin). Some of these drugs induce competence when applied at sub-MIC concentrations (Table S1). For ten of twelve stresses applied above their MIC (Supplementary Fig. 12) for 60 min, competent cells displayed increased survival compared to non-

**Fig. 2 | A self-induced subpopulation exists, and is increased upon antibiotic exposure, leading to propagation of competence throughout a population.**
**A** Schematic representation of experiment carried out to explore whether competence induction by exposure to sub-lethal concentrations of antibiotics follows an SI&P mode of transmission. **B** Competence levels tracked by RLU or transformation efficiency as shown in (**A**). Left panels, RLU OD[1] tracking competence at the populational level; right panels, transformation efficiency tracking competence at the individual cell level. The blue arrowed lines highlight a period with detection of low transformant levels (right graph) revealing a self-induced cell fraction, without detection of populational competence propagation (left graph). Bottom panels show that antibiotic exposure increases this subpopulation, leading to competence propagation. Values represent rate calculations of competence development. Vertical arrows represent time of addition of CSP or antibiotic. Decrease in

transformants and luminescence after ~200 min are explained by entry into stationary phase (Supplementary Fig. 7A). Individual data shown representative of duplicate repeats showing similar results. **C** Schematic representation of experiment mixing activated and non-activated cells to artificially create fractions of competent cells and compare $X_A$ times. Key as in Fig. 1B. **D** A critical artificial cell fraction can initiate competence propagation. Green shapes with colour gradient correspond to the $X_A$ values defined for varying ratios of activated/non-activated cells, framed by controls using 100% activated cells (dark green) and 100% non-activated cells (light green). Grey shapes correspond to the $X_A$ values extracted from the negative control experiments (activated cells replaced with medium exposed to CSP). Means and standard deviations are reported. Asterisks represent significance between transformation efficiencies. n.s. non-significant, $p > 0.05$; ***$p < 0.005$.

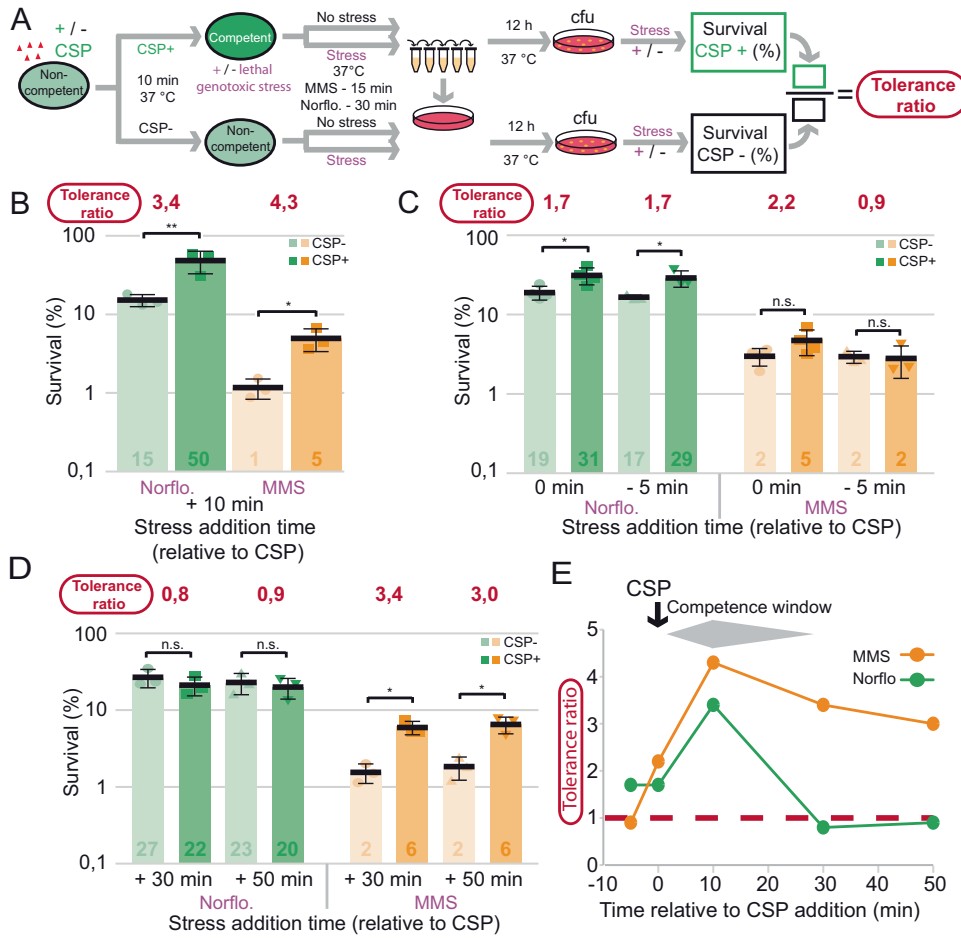

**Fig. 3 | Increased survival of CSP-induced competent cells exposed to MMS or Norflo. A** Simplified schematic of survival assay. Full experimental protocol in Supplementary Fig. 8. R1501 cells were induced to competence by CSP addition then exposed to stress. After 60 min, cells were serially diluted and plated. Tolerance ratios were calculated by first determining the survival percentage of stressed and non-stressed populations of competent and non-competent cells via colony counts and then dividing the survival percentage of competent cells by that of non-competent cells. **B** Survival of competent (dark) and non-competent (light) cells exposed to Norflo or MMS at +10 min relative to CSP addition. Cells exposed to Norflo for 30 min and MMS for 15 min. Survival percentages calculated compared to non-stressed cells. Tolerance ratios calculated as in panel A. Experiments were carried out in three biological replicates with means and standard deviations presented. Asterisks represent significance between transformation efficiencies.

*$p < 0.05$; ***$p < 0.005$. **C** Survival of competent and non-competent cells exposed to Norflo or MMS at −5 min or 0 min relative to CSP addition. Detailed assay description in Supplementary Fig. 8B, C. Colour scheme, exposure times and representations as in (**B**). Experiments were carried out in three biological replicates (shown by number of data points) with means and standard deviations presented. n.s. non-significant, $p > 0.05$; *$p < 0.05$; ***$p < 0.005$. **D** Survival of competent and non-competent cells exposed to Norflo or MMS at +30 min and +50 min relative to CSP addition, representing post-competent populations. Colour scheme, exposure time and representations as in (**B**). Experiments were carried out in three biological replicates with means and standard deviations presented. n.s. non-significant, $p > 0.05$; *$p < 0.05$. **E** Tolerance ratio plotted against time of stress relative to CSP for Norflo and MMS. Data taken from (**B**, **C**, **D**).

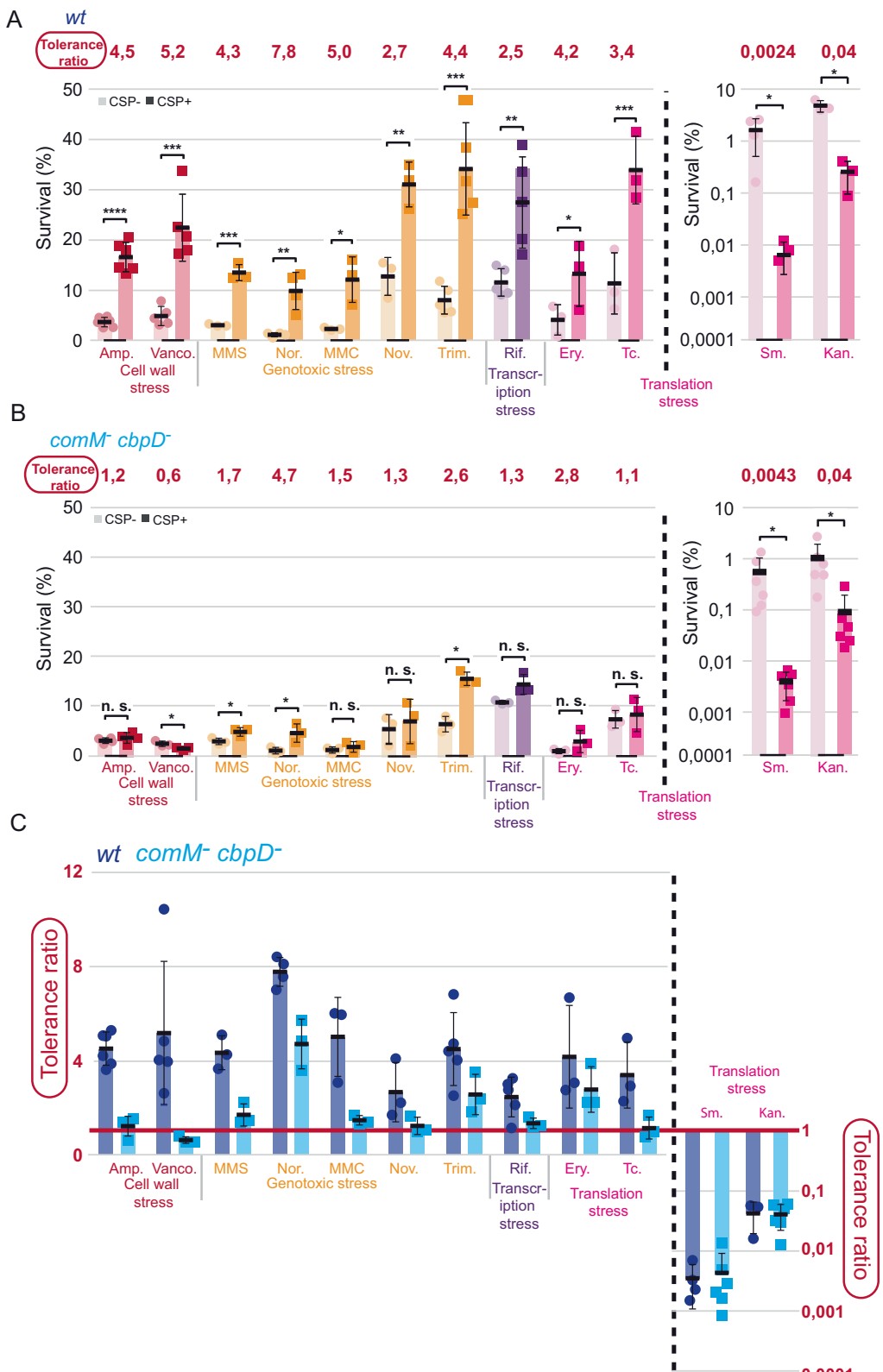

competent cells, with tolerance ratios between 2.5 and 7.8 (Fig. 4A). By stark contrast, competent cells displayed increased sensitivity to aminoglycosides (streptomycin and kanamycin), with tolerance ratios of 0.0024 and 0.04 respectively (Fig. 4A). Aminoglycosides were previously found to induce competence[33]. Since these effects were observed at a single time point, time course experiments were carried out for stresses for which competence increases survival (vancomycin)

or sensitivity (kanamycin), which showed that these effects were gradual between 20 and 60 min post-stress exposure (Supplementary Fig. 13A, B), as for norfloxacin (Supplementary Fig. 10D). In conclusion, competence modulates the survival of cells transiently exposed to a wide variety of lethal stresses, with a general benefit for competent cells, revealing phenotypic heterogeneity within the competent population.

**Fig. 4 | Competence modulates survival of pneumococci exposed to a wide range of stresses, with ComM key to increased survival. A** Survival of competent (dark) and non-competent (light) R3369 cells exposed to various stresses for 60 min starting at +10 min relative to CSP addition. Full experimental protocol in Supplementary Fig. 8A. Colours represent different types of stress. Black dotted line separates stresses where competence increased survival from those where competence increased susceptibility, plotted on a secondary y-axis. Tolerance ratios calculated as in Fig. 3A. Experiments were carried out in 3–6 biological replicates (denoted by number of data points) with means and standard deviations presented. Asterisks represent significance between survival of competent and non-competent cells. *$p < 0.05$; **$p < 0.01$; ***$p < 0.005$; ****$p < 0.001$. **B** Survival of competent (dark) and non-competent (light) *comM cbpD* cells (R4592) exposed to various stresses for 60 min starting at +10 min relative to CSP addition. Experimental procedures and representations as in (**A**). Experiments were carried out in 3–6 biological replicates (denoted by number of data points) with means and standard deviations presented. n.s. non-significant, $p > 0.05$; *$p < 0.05$. **C** Comparison of tolerance ratios of wildtype and *comM cbpD* cells, with values of 1 representing no difference between competent and non-competent cells, values >1 representing increased survival in competent cells and values <1 representing increased survival in non-competent cells. Ratios were calculated from 3 to 6 biological replicates (denoted by number of data points) with means and standard deviations presented.

## ComM is central to the improved tolerance to stress of competent cells

Increased tolerance has been associated with reduced growth and altered metabolism in other bacteria[48–52], which could play a role in competence-mediated tolerance increase. To test this hypothesis we inactivated *comM*, which promotes a division delay in competent cells[15], as well as *cbpD*, the gene encoding the fratricide hydrolase CbpD, to which ComM provides immunity[14]. We investigated the antibiotic panel with this *comM cbpD* mutant and the isogenic *comM+ cbpD+* parent strain. The absence of *comM* reduced the tolerance ratio for stresses where competence had increased survival, revealing ComM as key to increased survival during competence (Fig. 4B, C). In some cases, the tolerance ratio was reduced to ~1, suggesting that ComM alone was responsible for the increased survival during competence, but in others a tolerance ratio of >1 remained, suggesting other competence factors may also contribute. These results further highlight the heterogeneity present in a competent pneumococcal population. As a control, we also tested *comM* and *cbpD* single mutants. A *comM* mutant showed similar tolerance ratios to a *comM cbpD* double mutant, while a *cbpD* mutant showed similar tolerance ratios to wildtype. (Supplementary Fig. 13C, D, E). This showed that fratricide was not detected in these conditions, and demonstrated that CbpD plays no role in tolerance, with ComM alone responsible for this phenotype. Complementation of the absence of *comM* by the ectopic *CEP_R-comM* construct[18,53] restored the increased tolerance of competent cells to ampicillin or tetracycline, confirming ComM as a main actor of competence-induced tolerance to stress (Supplementary Fig. 14). In contrast, absence of *comM* (and *cbpD*) did not alter the tolerance ratio for the two aminoglycoside stresses where competence increased sensitivity (Fig. 4B, C). In conclusion, competence-mediated tolerance increase is mainly dependent on the early competence protein ComM.

## Competent-tolerant cells able to transform favour acquisition of gene cassettes over single nucleotide polymorphisms

We defined competent cells that tolerated stress as competent-tolerant and explored whether these cells retained the ability to transform under lethal stress. We used tDNA fragments with a selective antibiotic resistance provided via a heterologous gene cassette (HGC) or a single nucleotide polymorphism (SNP). We performed the survival assay with stress added 10 min after CSP, and exogenous transforming DNA (tDNA) was added at the same time as the stress, followed by 60 min exposure in liquid medium. Next, cells were diluted, plated and incubated for 80 min before transformant selection. This procedure allowed determination of transformation efficiency, along with survival and tolerance ratios (Fig. 5A and Supplementary Fig. 15). In HGC assays performed under five different stresses targeting different cell processes, the transformation frequency of tDNA conferring kanamycin resistance was reduced 3–8-fold, in comparison to a non-stressed population (Fig. 5B). In stark contrast, in the SNP assay using tDNA containing an *rpsL41* point mutation (A157C transversion) conferring streptomycin resistance[54,55], a significantly larger decrease in

transformation efficiency was observed, ranging from 43-156-fold (Fig. 5C). Thus, although the transformation efficiency of SNPs is 50-fold higher than that of HGC under non-stressed conditions, the former was much more affected by stress. This was also the case with another SNP in the *rpoB* gene (C1517T transition), conferring rifampicin resistance, *rif23*[54] (Fig. 5D). Thus, transformation efficiency was altered in competent-tolerant cells, as fewer cells were able to transform, further highlighting the adaptive heterogeneity of a competent population exposed to stress. The Hex mismatch repair (MMR) system interferes with transformation at the recombination step by recognising mismatches generated during heteroduplex formation. MMR, which is not induced during competence[10–12], recognises mismatches generated by transition (*rif23*) better than those generated by transversion (*rpsL41*)[56], explaining the ~4-fold deficit in transformation between these markers in non-stressed cells (Fig. 5C, D). We repeated the assay in an isogenic MMR- strain. As expected, the same transformation efficiency was observed for *rif23* and *rpsL41* in the absence of stress. Interestingly, we observed increased transformation efficiency of these two SNPs in competent-tolerant MMR- cells compared to MMR+ cells (expressed as MMR ratio; Fig. 5E). These results strongly suggest that MMR is hyper-active in competent-tolerant cells. In conclusion, competent-tolerant cells transform at low levels and favour HGC over SNP transfer due to increased MMR.

## Discussion

This work validates propagation as the general mode of pneumococcal competence transmission and reveals individual self-induced cells as the source of populational competence development. We call this mode SI&P (Fig. 2). Here, we present arguments showing that SI&P shapes competence as a populational health sensing mechanism equipped to discriminate incoming stress from false alarms. Competence relies on stochastic self-induction of a minority of cells in response to stress, while the majority have not sensed enough stress to self-induce, but are induced via propagation (Fig. 5F). In transformation assays, we detected one self-induced cell per $10^{-7}$, but competence propagation required a greater ratio (Fig. 2). The planktonic growth condition governs the probability of cell-to-cell contact and we deduce that the ratio of CSP-decorated self-induced cells to non-stressed cells must pass a threshold to propagate and convert the non-self-inducing cells (Fig. 2 and Supplementary Fig. 5F). The role of unphosphorylated ComE as a transcriptional repressor of competence should contribute as a mechanism guarding against false alarm, with the stoichiometry between the unphosphorylated transcriptional repressor ComE and the transcriptional activator ComE-P governing competence development[57–60]. Firstly, basally expressed ComE will dampen competence development at the cellular level, reducing the number of SI cells. Indeed, increasing basal expression of *comCDE* 10-fold inhibits competence, presumably due to increased levels of ComE repressor[59]. Overexpression of the *comAB* transporter genes removes this effect[59], probably by increasing export of CSP, which shifts the ComE/ComE-P ratio and allows self-induction. Secondly, ComE will increase the threshold of SI cells required to promote competence propagation.

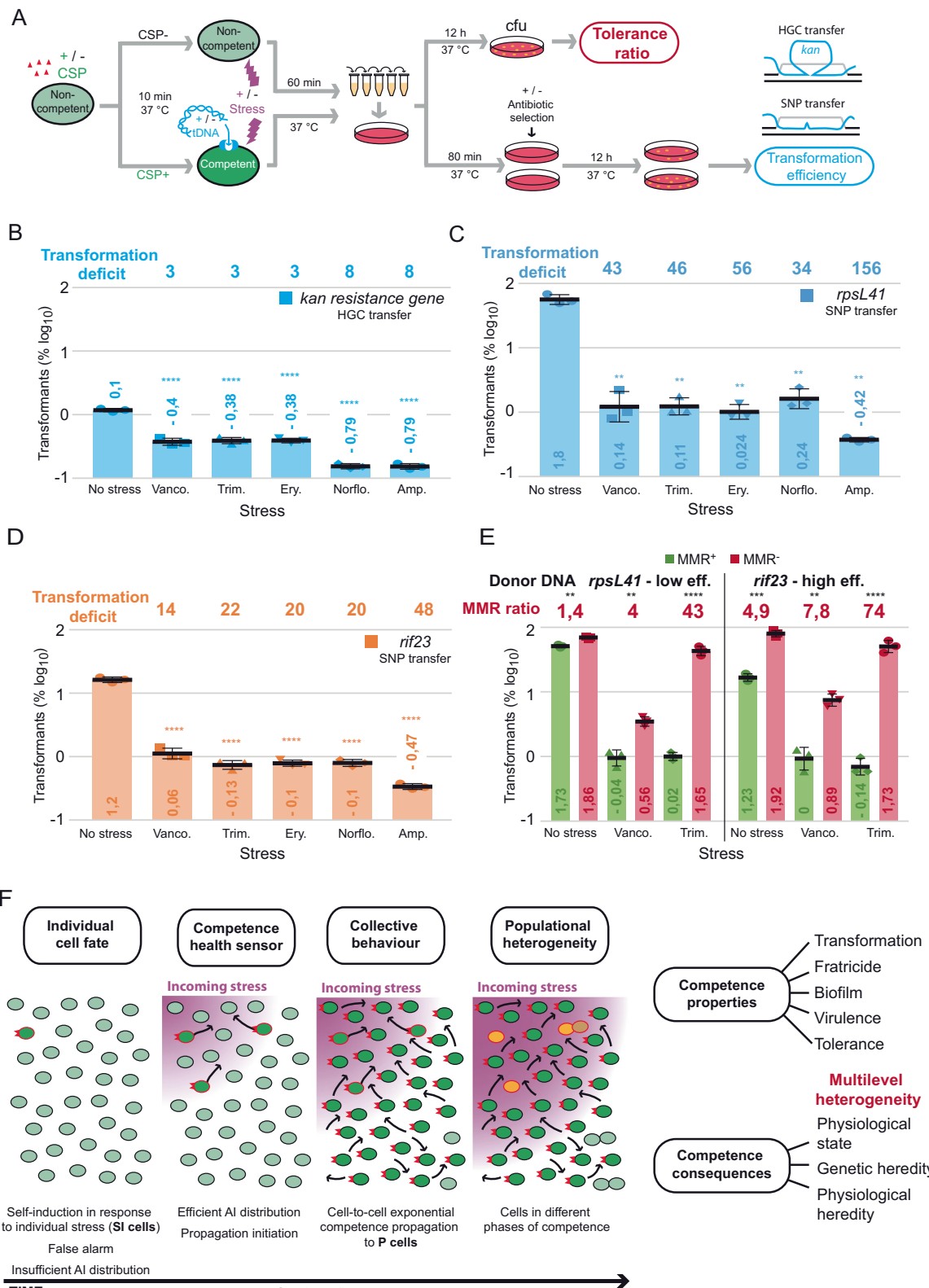

This prevents populational competence unless a threshold of self-induced cells is reached in stress conditions. Thus, the ComE/ComE-P ratio is central to defining a health sensor mechanism both at the individual and populational level. Once CSP distribution by cell-to-cell contact overcomes this protective mechanism, competence propagates. Such a mechanism benefits the population since it can avoid the cost of populational competence development via false alarm but

react efficiently to a stressful environment, rapidly converting the population to competence (Fig. 2). Propagation becomes exponential if the AI is distributed from one SI cell to several P cells (Fig. 1). This is in contrast with classical QS AI diffusion, where passive accumulation of AI leads to a linear response where all cells respond in synchrony once a threshold concentration of free AI is reached (Supplementary Fig. 1). Noise and stochasticity are key to the appearance of the SI fraction, but

**Fig. 5 | Tolerant cells favour integration of heterologous cassettes over point mutations by transformation. A** Transformation survival assay. Full experimental protocol in Supplementary Fig. 13. **B** Transformation efficiency of heterologous sequences in tolerant pneumococcal cells. Cells transformed with tDNA possessing a heterologous cassette conferring kanamycin resistance (*comFA::kan*). Experiments were carried out in triplicate with means and standard deviations presented. Asterisks represent significance between transformation efficiencies. ****$p$ < 0.0001. Tolerance ratios in Supplementary Table 2. **C** Transformation efficiency of *rpsL41* SNP in tolerant pneumococcal cells. Cells transformed with tDNA possessing SNP conferring streptomycin resistance (*rpsL41*). Experiments were carried out in triplicate with means and standard deviations presented. Asterisks represent significance between transformation efficiencies. ***$p$ < 0.005. Tolerance ratios in Supplementary Table 2. **D** Transformation efficiency of *rif23* SNP in tolerant pneumococcal cells. Cells transformed with tDNA possessing SNP conferring rifampicin resistance (*rif23*). Experiments were carried out in triplicate with means and standard deviations presented. Asterisks represent significance between transformation efficiencies. ****$p$ < 0.0001. Tolerance ratios in Supplementary Table 2. **E** Comparison of transformation efficiencies of point mutations (*rpsL41* and *rif23*) in MMR+ and MMR- cells. Experiments were carried out in triplicate with means and standard deviations presented. Asterisks represent significant

difference between MMR+ and MMR- populations (MMR ratio). **$p$ ˙ 0.001. Tolerance ratios in Supplementary Table 2. **F** Schematic representation of competence development. At low environmental stress levels, a small fraction of cells self-induces competence in response to individual cellular stresses such as metabolic breakdowns (SI cells), but does not distribute sufficient CSP to propagate competence. This prevents populational propagation resulting from false alarms. As stress increases, the fraction of SI cells reaches a threshold of CSP distribution able to convert neighbouring P cells to competence. This reveals competence as a populational health sensor. Competence can propagate exponentially rapidly through a population, ensuring that P cells not sufficiently stressed to self-induce become competent prior to stress exposure. This wave of competence generates populational heterogeneity, with a mixture of competent and post-competent cells. Competence promotes heterogeneity at multiple levels. This includes physiological state (altered stress tolerance), genetic heredity (transformation) and the physiological state within the next generation (lop-sided transmission of DprA), affecting subsequent competence waves. Cell identities as in Fig. 1B, with different shades of orange representing lop-sided DprA transmission to daughter cells. Purple wave represents incoming environmental stress. Black lines represent propagation of competence throughout the population.

additional environmental stresses increase this fraction, allowing the cell population to discriminate between noise and incoming stress (Fig. 2). SI has been observed in two QS systems in *Bacillus subtilis* as well as in *Pseudomonas aeruginosa*[61,62]. However, no populational propagation was observed in these systems. Altogether, these results reveal pneumococcal competence as a health sensor of the population, idling in the absence of danger but able to react extremely rapidly when danger is detected.

Pneumococcal competence not only acts as a populational health sensor, but also generates heterogeneity within a population as revealed by altered ability to tolerate transient exposure to lethal stresses (Figs. 3, 4)[23]. Competence increases tolerance to most tested stresses, except aminoglycosides, for which it dramatically increases sensitivity (Fig. 4). Internalisation of aminoglycosides is increased when the proton motive force (PMF) is high[63]. In addition, the PMF has been shown to be increased specifically during pneumococcal competence[64]. Thus, we suggest that this increases the cytoplasmic concentration of aminoglycosides, rendering competent cells more sensitive. Improved tolerance was observed for various lethal stresses targeting vital cellular functions (genome replication, translation, peptidoglycan synthesis), highlighting the wide-ranging protective effect provided by SI&P-driven competence. Improved tolerance against lethal drugs was also reported for competent *B. subtilis* cells, linked to ComGA-mediated growth arrest[51]. We show that improved pneumococcal tolerance also mainly depends on a transient growth delay mediated by the early competence protein ComM[15]. Nonetheless, competent cells lacking ComM still exhibit improved tolerance to some stresses (Fig. 4C), showing that other competence-induced effectors provide specific protection. Remarkably, there is no relation between increased tolerance to a particular stress and its ability to induce competence, suggesting that competence induction diversifies a population, providing the potential to tolerate a wide variety of stresses. Competence is known to provide transformation ability to all cells, potentially scattering a clonal population into a mosaic of genotypes. However, under lethal stress, competent-tolerant cells are biased towards transformation-mediated acquisition of HGCs over SNPs, due to increased MMR activity. Another level of heterogeneity comes via DprA, which accumulates at one cell pole to promote competence shut-off[32]. This results in lop-sided inheritance of DprA in daughter cells, which partly explains the poor coordination of secondary competence waves[28,29]. Overall, this shows that the competence health sensor fosters functional diversification via multilevel heterogeneity of a clonal population (genetic heredity via transformation, physiological state via altered tolerance, physiological

heredity via lop-sided distribution of DprA post-competence) (Figs. 1C and 5F), with some cells better suited to surviving various stresses.

Pneumococcal competence development not only promotes multilevel heterogeneity but self-induction is also driven by stochastic diversity between members of the population. This diversity generates a bimodality, with a minority of cells self-inducing and the majority remaining non-competent unless propagation conditions are met. This bimodality is a common feature of QS systems in other bacterial species, such as *B. subtilis*, where competence is induced in a fraction of cells based on the noisy expression of its master transcriptional regulator ComK[65,66]. *Streptococcus salivarius* also displays bimodality in competence development. The ComR transcriptional regulator[67] is negatively controlled by a two-component regulatory system, CovRS, with bimodality coming from heterogeneity in sensing environmental changes, without observed propagation[68]. A key difference compared to these two systems is that the bimodality of pneumococcal competence can lead to populational propagation. However, *Streptococcus thermophilus*, which also regulates competence via the two-component regulatory system ComRS, was shown to develop competence in a manner which could fit the SI&P model proposed here[69], although this remains to be investigated. Bimodal competence development has been suggested as a bet hedging strategy[68,70] where only a fraction of the population pays the energetic cost of competence but also reaps the potential benefits. This cannot be applied to pneumococcal competence due to its propagative and populational nature. Nonetheless, populational competence development, which initially depends on bimodality, scatters a population towards multi-level heterogeneity, fitting the definition of bet hedging. In contrast to both *B. subtilis* and *S. salivarius*, the pneumococcus is an important opportunistic human pathogen[71], exposed to environmental stresses such as host immunity and antibiotics during both commensal and pathogenic life. Our work shows that competence is tightly integrated into the pneumococcal lifestyle and key for the survival of this major human pathogen. Competence-mediated diversification of a clonal pneumococcal population may provide the best opportunity for survival in such environments. This idea is strengthened by the links between pneumococcal competence and virulence[18–20,22], and the suggestion that a single cell bottleneck is at the origin of pneumococcal bacteraemia[72].

This study reveals that pneumococcal competence responds to phenotypic heterogeneity and proceeds via a SI&P mode in the face of environmental stresses (Fig. 5F). Competence development drives multilevel heterogeneity, with one result being altered tolerance in the

face of transient lethal stress exposure. Competence can thus be framed as a global pneumococcal stress response, with transformation representing one of the multiple facets of this response. In conclusion, SI&P shapes pneumococcal competence as a health sensor tuned to scatter the clonal population towards multilevel diversification, employing a bet hedging strategy to maximise the survival potential of a pneumococcal population in response to environmental stresses such as antibiotics.

## Methods

### Bacterial strains, construction and transformation

*S. pneumoniae* strains and primers used in this study are described in Table S3. CSP-induced transformation[60] was performed using pre-competent cells treated at 37 °C for 10 min with synthetic CSP1 (100 ng mL$^{-1}$). After addition of tDNA, cells were incubated for 20 min at 30 °C. Transformants were selected by plating in 10 mL CAT-agar supplemented with 4% horse blood, incubating for 2 h at 37 °C for phenotypic expression and overlaid with 10 mL CAT-agar containing the appropriate antibiotic as followed: chloramphenicol (4.5 µg mL$^{-1}$), kanamycin (250 µg mL$^{-1}$), streptomycin (200 µg mL$^{-1}$), tetracycline (1 µg mL$^{-1}$). To generate strain R2287, R1501 was transformed with a mariner mutagenesis fragment generated by in vitro transposon mutagenesis of the *comFA* gene, generated using primer pair comFA3-comFA4, with the pR410 plasmid (KanR)[73]. A *comFA::kan* transformant with a kanamycin resistance cassette inserted in the *comFA* gene in a co-transcribed orientation was isolated by selection with kanamycin. To generate the R4428 strain, a kanamycin allele knock-out with a 7 bp deletion in the ORF was generated using the primer pairs MP259-MP2260 and MP261-MP262 to amplify two DNA fragments from strain R2737, followed by splicing by overlap extension (SOE) PCR using primer pair MP259-MP262. This DNA fragment was transformed into R2737[74], and a kanamycin-sensitive transformant was isolated. The presence of the *kan*$^S$ allele was validated by sequencing. To generate strain R4590, R3369 (*comC2D1*) was transformed with genomic DNA (gDNA) from strain R3967[15] and *comM::cat* transformants were isolated by selection with chloramphenicol. To generate strain R4591, R3369 (*comC2D1*) was transformed gDNA from strain R1620[75] and *cbpD::spc* transformants were isolated by selection with spectinomycin. To generate strain R4592, R4590 (*comC2D1, comM::cat*) was transformed with gDNA from strain R1620[75] and *cbpD::spc* transformants were isolated by selection with spectinomycin. To generate strain R5218, R4590 (*comC2D1, comM::cat*) was transformed with gDNA from strain R3957[15] and *CEP$_R$-comM* transformants were isolated by selection with kanamycin. To generate TD277, D39$_{Tlse}$ was transformed with gDNA from strain R3316[36], with P$_{ssbB}$::*luc* transformants isolated by selection with chloramphenicol. To generate strain TD288, D39V[76] was transformed with gDNA from strain R3316[36], with P$_{ssbB}$::*luc* transformants isolated by selection with chloramphenicol.

### Spontaneous competence detection using luciferase gene reporter

Competence development was monitored using P$_{ssbB}$::*luc*[40]. In these experiments, C + Y medium at different initial pH ranges was used to control the ability of cells to spontaneously develop competence[39], with acidic pH 6.8–7.0 being non-permissive to competence and alkaline pH 7.6–7.9 permissive. This nomenclature will be used throughout this manuscript. Briefly, cells were grown for several generations to mid-log phase in non-permissive medium, washed and stored at −70 °C in fresh medium supplemented with 20% glycerol. This cell stock was used to generate varying inoculum sizes in permissive medium in a clear-bottomed 96-well white NBS microplate (Corning). Cells were grown at 37 °C in a Varioskan Flash (Thermo 399 Electron Corporation) luminometer and relative luminescence units (RLU) and OD$_{492}$ values were recorded throughout incubation. To calculate the X$_A$ and X$_B$ values, the theoretical regression of each slope

before and during the first wave of competence propagation was obtained. The intersection of these slopes defines the X$_A$ value (in minutes), and the rate of the exponential slope during propagation defines the X$_B$ value. In some cases during the X$_A$ period, stochastic heterogeneity means that the RLU values are below the level of sensitivity of the luminometer and provide negative values when blank values are subtracted. We were unable to calculate RLU/OD values from these values, which were removed from the analysis. This results in occasional gaps between data points during the X$_A$ period.

### Spontaneous competence detection using transformation assays

14 mL of permissive medium was inoculated with two different cell densities at 37 °C to provide high and moderate X$_B$ competence development rates[39]. During growth, every 10 min, 100 µL of cells were tested for transformation after 10- or 20-min exposure to tDNA and 300 µL were used to measure photon emission and OD$_{492}$. A control was generated by the addition of synthetic CSP after 70 min of growth to obtain an artificial fully coordinated competent population[45,77]. Each transformation assay was carried out in 100 µL aliquots of cell culture. tDNAs used for measuring transformation efficiency were the purified 3434 bp PCR fragment generated with the primer pair MB117-MB120 containing a SNP conferring streptomycin resistance[78] or the purified 3238 bp PCR fragment containing the wildtype kanamycin resistance allele generated by the primer pair MP259-MP262. Both tDNA PCR fragments were used at 200 ng mL$^{-1}$. The streptomycin resistance allele differs by a single nucleotide from the sensitive wildtype allele. The kanamycin sensitive allele differs from the kanamycin resistant allele by a 7-nucleotide deletion in the open reading frame. Both mutations are central within the PCR fragments used. Without transformation, no spontaneous kanamycin resistant clones were detected in R4428. Cells are incubated with tDNA at 30 °C during 10 or 20 min followed by addition of 20 µg mL$^{-1}$ DNase I (Sigma Aldrich) with MgCl$_2$ at 10 mM for a further 10 min at 37 °C before plating. Selection of the transformants was done using streptomycin at 200 µg mL$^{-1}$ or kanamycin at 250 µg mL$^{-1}$. Controls with no tDNA or with concomitant addition of tDNA and DNase I were repeated several times and gave no detectable transformants.

### Spontaneous competence detection by visualisation of DNA binding using microscopy

Analysis of DNA binding was performed in a virulent *endA* mutant (TD290), to favour accumulation of transforming DNA at the surface of competent cells. In wildtype, *endA*$^+$ cells, surface-bound DNA is immediately internalised into the cytosol or degraded, which makes surface-bound DNA accumulation hard to visualise[41]. Precultures were inoculated at two cell densities ($8 \times 10^{-3}$ and $8 \times 10^{-4}$ OD$_{550}$) in permissive medium and allowed to grow, with samples taken at various time points through growth to determine DNA binding. X$_A$ detection was carried out by using P$_{ssbB}$::*luc*. The positive control with CSP addition on the low cell density inoculum was carried out at the X$_A$ time. For each time point along growth an aliquot was incubated for 5 min with 10 ng of a 285 bp DNA fragment labelled with a Cy3 fluorophore at its 5' extremity, amplified using primer pair OCN75-76[41]. Cells were pelleted ($3000 \times g$, 3 min), washed twice in 500 µL C + Y, and resuspended in 20−50 µL C + Y medium before microscopy. 2 µL of this suspension was spotted on a microscope slide containing a slab of 1.2% C + Y agarose[79]. Phase contrast and fluorescence microscopy were performed with an automated inverted epifluorescence microscope Nikon Ti-E/B, a phase contrast objective (CFI Plan Apo Lambda DM 100X, NA1.45), a Semrock filter set for Cy3 (Ex: 531BP40; DM: 562; Em: 593BP40), a LED light source (Spectra X Light Engine, Lumencor), and a sCMOS camera (Neo sCMOS, Andor). Images were captured and processed using the Nis-Elements AR software (Nikon). Cy3 fluorescence images were false coloured red and overlaid on phase contrast

images. Overlaid images were further analysed to quantify the number of cells bound with Cy3-labelled DNA. Single cells were first detected using the threshold command from Nis Elements and cells bound or not to DNA were manually classified using the taxonomy tool. Data representative of three independent repeats.

## Cell mixing experiment

A stock of the R4428 strain grown in non-permissive medium conserved in glycerol at −70 °C was used to inoculate non-permissive medium at 37 °C at $OD_{550}$ 0.004. Cells were grown to $OD_{550}$ 0.06 and growth was stopped on ice. Cells were washed by centrifugation and resuspended in permissive medium at 4 °C. A cell aliquot and a sterile medium aliquot without cells (negative control) were exposed to 25 ng mL$^{-1}$ of CSP during 1 min at 37 °C then placed on ice. Each assay was washed 3 times by centrifugation with the same volume of sterile permissive medium at 4 °C. Different dilutions of both cell aliquots and sterile medium aliquots were mixed with a constant number of cells not exposed to CSP and loaded directly on a clear-bottomed 96-well white NBS micro plate (Corning). The microplate was incubated at 37 °C immediately and the Relative luminescence units (RLU) and $OD_{492}$ recorded every minute in a Varioskan Flash (Thermo 399 Electron Corporation) luminometer. cfu were counted to obtain the ratio in the assays between cells exposed (or not) to CSP.

## MIC measurements and survival assays

To determine the MIC for each antibiotic used, pneumococcal cells were inoculated in C + Y medium (pH 7.9) at $OD_{550}$ 0.08, and exposed to a concentration gradient of each antibiotic then grown for 400 min at 37 °C. $OD_{492}$ readings were taken every 5 min, and the MIC was determined as the lowest concentration showing no growth during the experiment (Supplementary Figs. 10, 12, Supplementary Fig. 1). Strains used for these experiments were unable to spontaneously induce competence since they either possessed a *comC0* mutation[11] or the combination of the non-compatible alleles *comC$_2$* and *comD$_1$*[18]. The experiments are summarised in Supplementary Fig. 11. Strains were grown in permissive medium to $OD_{550}$ 0.2, washed and concentrated to $OD_{550}$ 0.4 in C + Y + 20% glycerol and stored at −70 °C. Cells used to inoculate were either defrosted or pre-inoculated and grown exponentially to $OD_{550}$ 0.1, followed by dilution in permissive medium at OD 0.004 and incubation at 37 °C. Cells were grown during 20 min at 37 °C and divided into two samples, one with 100 ng mL$^{-1}$ CSP, the other sample without. Each sample was incubated for a further 10 min and again divided into two samples, one exposed to stress and one not. After 60 min incubation at 37 °C, cells were serially diluted, plated at appropriate dilutions on CAT-agar supplemented with 4% horse blood (two plates per dilution) and incubated at 37 °C overnight. 60 min was chosen as a stress exposure time since this was close to the MDK$_{99}$ of norfloxacin (Supplementary Fig. 10). Other timings used are as described in the text. For time-course experiments, culture volumes were scaled up and cfu were measured for each time point. Stress concentrations used were at least twice the MIC, and in some cases more, in an attempt to reach 99% of cells killed in the experimental set-up. These values can be found in Supplementary Table 1. Comparison of total cfu between cultures exposed or not to stress allowed calculation of survival as a percentage, with comparison of competent and non-competent percentages (tolerance ratio) showing the effect of competence on survival. Each experiment was repeated at least 3 times, as described in each Figure legend. Mean and standard deviations were calculated for each experiment, and an unpaired Student's t-test was used to determine the significance of the difference in survival between populations.

## Transformation survival assays

Transformation survival assays were carried out based on survival assays used in this study, but with modifications as shown in Supplementary Fig. 15. Briefly, precultured cells were rediluted in fresh medium at $OD_{550}$ 0.004 and incubated for 20 min at 37 °C before being split into two samples, one with 100 ng mL$^{-1}$ CSP, the other sample without CSP. Samples were then incubated for 10 min at 37 °C. Non-competent cells were then split in two, one exposed to stress and one not. Competent cells were split into four, with and without stress and tDNA. After 60 min incubation at 37 °C, cells were serially diluted and plated at appropriate dilutions to detect total cfu and transformant cfu on CAT-agar supplemented with 4% horse blood (two plates per dilution). Plates were incubated at 37 °C for 80 min to allow phenotypic expression of transformed phenotypes, before a second layer of medium containing appropriate antibiotic was added to transformant cfu plates for selection. Plates were then incubated overnight at 37 °C. Transformation efficiencies were calculated by comparing total cfu with transformant cfu. Stress concentrations used can be found in Table S1. Three DNA fragments were used for transformation, an *rpsL41* point mutant conferring streptomycin resistance, a *rif23* point mutant conferring rifampicin resistance and a *comFA::kan* cassette conferring kanamycin resistance. A 3434 bp DNA fragment containing *rpsL41* was amplified from strain R304 using primer pair MB117-MB120. A 4195 bp DNA fragment containing the *rif23* point mutation was amplified from strain R304 using primer pair MB137-MB138. A 4931 bp DNA fragment containing *comFA::kan*[3C] was amplified from strain R2287 using primer pair CJ339-CJ356. Antibiotics used for selection were streptomycin (200 µg mL$^{-1}$), rifampicin (2 µg mL$^{-1}$) and kanamycin (250 µg mL$^{-1}$). Each experiment was repeated in triplicate, with means and standard deviations calculated and a Student's t-test used to determine the significance of the difference in transformation efficiencies.

## Statistical analyses

Statistical analyses carried out in this study were unpaired Student's *t* tests, used to compare two individual data sets in GraphPad Prism. Two-tailed *p* values calculated in this way are detailed in each appropriate Figure legend.

## Reporting summary

Further information on research design is available in the Nature Portfolio Reporting Summary linked to this article.

## Data availability

The authors declare that the data supporting the findings of this study are available within the paper and its supplementary information files. Source data is provided with this paper. Source data are provided with this paper.

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

## Acknowledgements

We thank Jan-Willem Veening for providing us with strains and medium, allowing us to understand the differences between the two studies. This work was funded by the Centre National de la Recherche Scientifique, University Paul Sabatier and the Agence Nationale de la Recherche (grants ANR-10-BLAN-1331 and ANR-17-CE13-0031).

## Author contributions

M.P., C.H.G.J., N.C. and P.P. conceived the study. M.P., C.H.G.J., A.-L.S. A.B. and D.D.L. carried out experiments. M.P., C.H.G.J., N.C. and P.P. analysed the data. M.P., C.H.G.J. and P.P. wrote the paper.

## Competing interests

The authors declare no competing interests.
