## [Peer Review file · Nature Communications]

REVIEWER COMMENTS

Reviewer #1 (Remarks to the Author):

The manuscript by Prudhomme et al provides important insights into the development and propagation of cell-to-cell signaling during the development of competence in *Streptococcus pneumoniae*. There are five key findings/advances presented: 1. an existing debate in the field, whether the CSP signaling occurs through a culture-wide diffusion with synchronized response OR, if signaling is propagated at a localized (possibly contact-mediated) cell-to-cell level. The authors are rigorous in using multiple strain types, luminescence and fluorescence imaging and modeling to present a very strong argument that the latter model is correct. 2. Self-induction of competence occurs in a small subset of the population, likely by a stochastic process, and the probability of self-induction is influenced (increased mostly) by stress-inducing conditions. Antibiotics are the stressors tested in this manuscript and interestingly, compounds with different mechanisms of action affect induction differently. 3. Induction of competence leads generally to enhanced numbers of tolerant cells in the population, and again, induction is seen to have variable effects in terms of tolerance to different antibiotic classes—tolerance is even negatively impacted for aminoglycosides, for instance. 4. Surprisingly, competent cells surviving antibiotic challenge were found to have a disproportionately lower ability to integrate single-nucleotide-mismatched transforming DNA (tDNA) as compared to gene-cassette tDNA. This is explained by the Hex mismatch repair (MMR) system interfering, possibly due to up-regulation of MMR. 5. Finally, given the ranges of phenotypic outcomes both in response to stressors and tolerance to antibiotics, the authors characterize competence induction as driving multilevel heterogeneity. Overall, the paper is thorough, clearly written, and supported by elegant, rich figures. The work addresses important concepts in bacterial communication pertaining to signal initiation and propagation, and phenotypic outcomes of competence induction beyond transformation.

Comments/suggestions:

1. The primary hallmark supporting cell-to-cell contact-mediated propagation of competence development is the rate dependency on culture density. The theoretical model presented in Fig. 1D provides a clear image of the model, but would it be even more beneficial if a model of the alternative hypothesis (diffusion) be included in the figure? Regarding the propagation model, the prior publication (Prudhomme, et al, 2016 PLoS Genetics) provides many important aspects of the mechanism such as CSP association with the cell envelope; however, would a similar pattern in expanding competence be seen if CSP signaling relied on diffusion from the cell but the half-life of the released peptide was short? My comment is not meant to challenge the findings, but only to suggest that maybe it should be less adamant that cell-to-cell contact is the mechanism. Simply stating “the observations are consistent with a model whereby cell-contact (or a short-lived signal) mediates propagation”, would suffice.

2. I suggest that 'multilevel' heterogeneity be defined earlier in the text. It would help the reader to understand what is meant by 'multiple levels', presumably different phenotypes?
3. At lines 204 and 323 it is stated that only a minority of cells are stressed and the remaining majority are non-stressed. I suspect this is not accurate, and without direct measurement this shouldn't be stated. Would it be more accurate to state that under stressful conditions only a minority of cells respond by inducing competence?
4. Could the authors speculate further on the threshold phenomenon, that a critical number of cells need to be self-induced before propagation ensues? Allusion to the level of phosphorylation of ComE:ComE~P stoichiometry is an interesting possibility (but a citation for the paper is not listed in the references). Does this refer to the stoichiometric ratio within self-induced cells or in non-competent cells? Could other possibilities exist, perhaps the half-life of CSP influences the likelihood of propagation, and/or the half-life of CSP is influenced by stress?
5. Does the biphasic SI&P model enhance or diminish the likelihood of fratricide occurring than if an 'all-at-once' response was the mode of signaling? It seems there are more opportunities for fratricide during propagation of low-density populations. I wonder if measurement of this has been attempted?
6. Do the authors have any thoughts as to why MMS tolerance lasts longer than that seen for norflo?
7. Since tolerance studies were completed for the comM mutant (Fig S12), why not include these in main text instead of the comM cpbB mutant?
8. Lines 336-7 state that propagation leads to exponential responses while AI diffusion leads to a linear response. Wouldn't the inherent positive feedback of CSP production also lead to an exponential response for the diffusion model? Would 'chain reaction' be a helpful way to describe the phenomenon?
9. Is anything in the literature known about MMR induction during competence development? Are any of the MMR genes within the ComE or ComX regulon?

Minor

1. Line 186, the term 'artificially self-induced' doesn't seem appropriate. Aren't cells either artificially induced OR self-induced?
2. Line 215, would help to indicate in the text the strain names that cannot naturally induce competence. Tables were not included in the supplemental information. Not a major issue for the review but be sure to include in final versions.
3. Line 271, "...in other bacteria could play..." seems like the word 'and' is needed.
4. Line 273, instead of 'screen' and 'isogenic wildtype', perhaps '...repeated the antibiotic panel with the isogenic strain.'
5. Just want to be sure strain R4950 construction is described in the final version.

6. Wondering why some graph lines in figure S2B (and others) are not connected.

Reviewer #2 (Remarks to the Author):

In this work, the author revisits the mechanism of competence induction and role of competence in pneumococci. Through a series of experiments, the authors conclude that competence is a “population health sensor” in pneumococci, in which self-activation of competence in a small fraction of cells in the population in turn leads to propagation of competence throughout a population via cell-cell contacts - in a stress-related manner. It is suggested that this a mechanism to promote heterogeneity and diversity in the population, providing protection towards different stressors, as the competent state is also shown to alter the sensitivity of the population to antimicrobial agents. In line with previous reports, competence increase the tolerance to some antimicrobials, and interestingly it is here shown that this tolerance is modulated by the timing of stress induction and competence induction - and it is shown that the ComM protein is essential for this protection. The results also show that transformation efficiencies and properties are altered in such tolerant cells.

The manuscript suggests a new way to view the function of pneumococcal competence. The data overall seem robust and the paper includes a large number of thorough and rather complicated experiments. Although I find the results and models proposed interesting, some of the statements and conclusions should be moderated (see below). In addition, several points needs clarification.

Main points:

1. L138-9 and Supplementary information. The manuscript starts with an investigation of the differences in results and conclusions between the authors’ previous work (concluding that competence is propagated through cell-cell contact) and another study (Moreno-Gamez et al., concluding that competence is activate in the population at a certain AI concentration without the need a cell-cell contact). A number of experiments comparing the media, strains and luminometers are performed, and the authors conclude the results differ due to differences in reporter constructs and sensitivity of luminometers.

a. L939 – 942. Were the two D39-strains genotypically identical (except for the different reporter constructs)? It is not clear to me whether the two reporter constructs were compared in the exact same strain background (I cannot find any strain list or methodology describing this).

- b. Fig. S1. Was the growth (OD) of the strains monitored during this experiment? Please include information about the growth.
- c. Explain precisely how X_b and X_a was calculated in the Methods-section or first time it appears in the figure.
- d. Comparison of luminometers. What is meant by “reference luminometer”?
- e. Line 951: please elaborate on how these results can “confirm that propagation is the general mechanism”.
- f. What is not addressed clearly here is the cell-cell contact hypothesis. In Moreno-Gamez et al. it is shown (using microscopy) that competence could spread between cell/colonies without the need of cell-cell contact. Have such an experiment been tested here? How can it be excluded that competence propagation without cell-cell contact could play a role under certain conditions as well?

2. The term “population health sensor”

- a. Throughout the paper it is stated that the results “revealed” competence “population health sensor”. I think this is an interesting model based on the data shown here, but some of these statements should be moderated. It is not yet known whether such a mechanism will work under more in-vivo like conditions. I also find “health” sensor is also too broad, as competence only will develop as response to some stressors. And isn't it rather a stress sensor than a health sensor?
- b. Discuss/explain how stress mechanistically will affect the SI & P model proposed. Will it affect both the self-induction rate and the propagation, or both? Will all the stressors work through modification of the ComE-P pool? How does this proposed mechanism relate to the previously published mechanism for competence regulation by antibiotics (eg. PMID: 21933920, PMID: 32130952, PMID: 24725406).

3. Fig. 2B (upper panel). No error bars are shown. Include a measure on the variability between experiment related to the detection of this transformable subpopulation. There also appear to be some timepoints where no transformants were detected. Were this the same in all replicate experiments?

4. Line 200. How does these results “further validate” the mechanism proposed? Please elaborate.

5. Fig. 3A. “Tolerance ratios were calculated by comparing cfu of stressed and non-stressed populations.” Please be more specific in the definition of tolerance ratios.

6. Tables S1 – S3 are lacking.

7. Fig. 3 and Fig. 4. Specify how the drug concentrations (relative to MIC) was standardized for the drugs.

8. L348. The results show that competence increases tolerance to most drugs, except aminoglycosides streptomycin and kanamycin, but there is no further explanation about this. Can the authors (in the discussion section) speculate about potential reasons for this difference? (both in terms of mechanism and if there is any potential ecological relevance to this).

9. Fig. 4BC, Fig. S12 and related text related to the comM mutants

a. Complementation assays (eg. using inducible comM expression) should be performed for at least some of the antibiotics to confirm that the phenotypes can be complemented.

b. L356. It has previously been shown that a comM mutants have a cell division defect (Berge 2017). Here, the authors show that drug tolerance is reduced in the absence of comM. Can the author really conclude that the effect of comM related to antibiotic tolerance is “specific”, or could it be because the cells are already “weakened” in the absence of comM? Previous literature on the function of ComM should be discussed in the light of the new results obtained here.

10. Do the authors think that the mechanism proposed here would occur also in other peptide-based AI two/three-component systems, or is this a mechanism specific to pneumococcal competence.

Minor points.

L331. “health” instead of “heath”

L335-337. Provide reference or describe how AI diffusion leads to linear response.

L350. What does “...” mean?

P13. The Engelmoer&Rozen paper (doi: 10.1111/j.1558-5646.2011.01402.x) should also be cited in the discussion.

Fig. 5. Explain the term “Self induction in response to breakdown” and “Next generation scattering”.

L394 and L399. The same claim on pneumococcal competence as a health sensor is repeated twice within 5 lines.

Line 191-192 and Fig. 2D. It would be useful to see examples of plotted data from this experiments used to calculate the X_a .

REVIEWER COMMENTS

Reviewer #1 (Remarks to the Author):

The manuscript by Prudhomme et al provides important insights into the development and propagation of cell-to-cell signaling during the development of competence in *Streptococcus pneumoniae*. There are five key findings/advances presented: 1. an existing debate in the field, whether the CSP signaling occurs through a culture-wide diffusion with synchronized response OR, if signaling is propagated at a localized (possibly contact-mediated) cell-to-cell level. The authors are rigorous in using multiple strain types, luminescence and fluorescence imaging and modeling to present a very strong argument that the latter model is correct. 2. Self-induction of competence occurs in a small subset of the population, likely by a stochastic process, and the probability of self-induction is influenced (increased mostly) by stress-inducing conditions. Antibiotics are the stressors tested in this manuscript and interestingly, compounds with different mechanisms of action affect induction differently. 3. Induction of competence leads generally to enhanced numbers of tolerant cells in the population, and again, induction is seen to have variable effects in terms of tolerance to different antibiotic classes—tolerance is even negatively impacted for aminoglycosides, for instance. 4. Surprisingly, competent cells surviving antibiotic challenge were found to have a disproportionately lower ability to integrate single-nucleotide-mismatched transforming DNA (tDNA) as compared to gene-cassette tDNA. This is explained by the Hex mismatch repair (MMR) system interfering, possibly due to up-regulation of MMR. 5. Finally, given the ranges of phenotypic outcomes both in response to stressors and tolerance to antibiotics, the authors characterize competence induction as driving multilevel heterogeneity. Overall, the paper is thorough, clearly written, and supported by elegant, rich figures. The work addresses important concepts in bacterial communication pertaining to signal initiation and propagation, and phenotypic outcomes of competence induction beyond transformation.

We thank the reviewer for their kind words regarding our work, and for their pertinent proposals that have undoubtedly helped us to improve our article. We have answered all concerns raised point by point in blue below.

Comments/suggestions:

1. The primary hallmark supporting cell-to-cell contact-mediated propagation of competence development is the rate dependency on culture density. The theoretical model presented in Fig. 1D provides a clear image of the model, but would it be even more beneficial if a model of the alternative hypothesis (diffusion) be included in the figure?

We have added the equivalent figures of Figure 1BCD for the alternative model as Figure S1, and adjusted the Figure list in consequence.

Regarding the propagation model, the prior publication (Prudhomme, et al, 2016 PLoS Genetics) provides many important aspects of the mechanism such as CSP association with the cell envelope; however, would a similar pattern in expanding competence be seen if CSP signaling relied on diffusion from the cell but the half-life of the released peptide was short? My comment is not meant to challenge the findings, but only to suggest that maybe it should be less adamant that cell-to-cell contact is the mechanism. Simply stating “the observations are consistent with a model whereby cell-contact (or a short-lived signal) mediates propagation”, would suffice.

The reviewer is right, we cannot exclude such a hypothesis, however our previous attempts (Prudhomme et al., 2016) to detect CSP in the supernatant during the first wave of competence failed. Moreover, assays with membrane cutoffs showed that synthetic CSP crossed the membrane

but CSP produced by cells did not during the first wave of competence. In addition, unless CSP is actively degraded, it appears to be very stable in this medium. Without other experimental evidence, we favor the simplest hypothesis that cell-to-cell contact facilitates CSP transfer.

2. I suggest that 'multilevel' heterogeneity be defined earlier in the text. It would help the reader to understand what is meant by 'multiple levels', presumably different phenotypes?

We understand the reviewers point here as we refer several times to multilevel heterogeneity before truly defining it, which can lead to confusion. However, defining it earlier would also be confusing, as we would not have given all the arguments to explain it. As a result, we have removed reference to 'multilevel' until it is properly defined, and referred only to heterogeneity. Also, we have added more detail to our definition of multilevel heterogeneity, which now reads:

'Overall, this shows that the competence health sensor fosters functional diversification via multilevel heterogeneity of a clonal population (**genetic heredity via transformation, physiological state via altered tolerance, physiological heredity via lop-sided distribution of DprA post-competence**) (Figure 1C and 5F), with some cells better suited to surviving various stresses.'

3. At lines 204 and 323 it is stated that only a minority of cells are stressed and the remaining majority are non-stressed. I suspect this is not accurate, and without direct measurement this shouldn't be stated. Would it be more accurate to state that under stressful conditions only a minority of cells respond by inducing competence?

We agree with the reviewer that 'non-stressed' is not fully accurate. What we really mean is cells that have not perceived sufficient stress to promote self-induction of competence, whatever the levels of stress perceived. We have reworded these parts and replaced 'stressed' and 'non-stressed' cells with 'self-inducing' (SI cells) and non-self-inducing cells that can be induced by propagation (P cells).

4. Could the authors speculate further on the threshold phenomenon, that a critical number of cells need to be self-induced before propagation ensues? Allusion to the level of phosphorylation of ComE:ComE~P stoichiometry is an interesting possibility (but a citation for the paper is not listed in the references). Does this refer to the stoichiometric ratio within self-induced cells or in non-competent cells? Could other possibilities exist, perhaps the half-life of CSP influences the likelihood of propagation, and/or the half-life of CSP is influenced by stress?

Firstly, regarding the citation, we thank the author for highlighting this. The Martin et al., 2013 reference is present in the text (line 331) but not in a format referenced in the reference list. This has been corrected, as well as for other references in the same situation line 340.

Regarding the reviewer's question, we postulate that unphosphorylated ComE plays a role in both self-induced and non-competent cells, based on the fact that unphosphorylated ComE is known to be a transcriptional repressor of early competence genes (Martin et al., 2013). We suggest firstly that the presence of unphosphorylated ComE will dampen self-induction of competence in individual cells, reducing the number of self-induced cells in the growing population. This is based on data showing that cells displaying 10-fold increased basal *comCDE* expression can no longer self-induce competence in permissive medium, a defect that we attribute to an excess of unphosphorylated ComE resulting from an imbalanced *comABCDE* positive feedback loop (Guiral et al., 2006). This hypothesis is supported by over-expression of *comAB*, which was found to rescue the positive ComABCDE feedback loop, showing that CSP export by ComAB is a limiting process (Claverys and

Havarstein, 2002; Martin et al., 2000), and that an increase in CSP export shifts the ComE/ComE~P ratio towards allowing self-induction. Secondly, unphosphorylated ComE in P cells will increase the threshold of CSP distribution by self-induced cells required to promote competence propagation. This prevents populational competence with a low ratio of SI cells unless a threshold of SI cells is reached due to increased stress levels. Thus, the ComE/ComE~P ratio is central to the populational health sensor. We have adjusted the text to elaborate on this point as follows:

'The role of unphosphorylated ComE as a transcriptional repressor of competence should contribute as a mechanism guarding against false alarm, with the stoichiometry between the unphosphorylated repressor ComE and the transcriptional activator ComE~P governing competence development⁵⁶⁻⁵⁹. Firstly, basally expressed ComE will dampen competence development at the individual cell level, reducing the number of self-induced cells in the population. Indeed, increasing basal expression of *comCDE* 10-fold inhibits competence, presumably due to increased levels of ComE repressor⁵⁸. Overexpression of the *comAB* transporter genes counteracts this effect⁵⁸, probably by increasing export of CSP, which shifts the ComE/ComE~P ratio to allow self-induction. Secondly, a high ComE/ComE~P ratio will increase the threshold of self-induced cells required to promote competence propagation. This prevents populational competence unless a threshold of self-induced cells is reached in increasing stress conditions. Thus, the ComE/ComE~P ratio is central to defining a health sensor mechanism both at the individual and populational level.'

5. Does the biphasic SI&P model enhance or diminish the likelihood of fratricide occurring than if an 'all-at-once' response was the mode of signaling? It seems there are more opportunities for fratricide during propagation of low-density populations. I wonder if measurement of this has been attempted?

We have not carried out such experiments. Fratricide depends on CbpD in liquid cultures, with ComM providing immunity. However, fratricide has mostly been observed in high density cultures, while initial competence development occurs at low cell density. This is an interesting question, which we have already addressed in the discussion of our previous study, referring to a competition between competence induction and fratricide upon cell contact (Prudhomme et al., 2016). As a result, we feel it is out of the scope of this study and represents an independent project.

6. Do the authors have any thoughts as to why MMS tolerance lasts longer than that seen for norflo?

MMS and norfloxacin damage DNA differently, and the mechanisms involved in repairing these damages differ. Also, the distinct effectors involved in these mechanisms may act differently post-competence. One possibility is that such effectors have been overexpressed during competence and distributed into post-competent cells, rendering them more efficient in repairing genomic damages compared to non-competent cells. Another possibility could be that competent cells have expressed a way to efficiently expel MMS, a protection mechanism that could also be inherited in daughter cells and gradually decreased over generations. In addition, the antibiotic stress may induce other regulons which may impact tolerance. These important and interesting questions form part of future studies that we will undertake to explore the heterogeneity of competence-induced tolerance.

7. Since tolerance studies were completed for the *comM* mutant (Fig S12), why not include these in main text instead of the *comM* *cpbB* mutant?

Our rationale was that since *comM* mutants are lacking the immunity protein for fratricide, they could be sensitive to suicide by production of the fratricide hydrolase CbpD. To remove this

possibility, we used the double mutant. However, we carried out experiments in the single mutants as controls and saw similar results (showing that fratricide is not active in these conditions). We adjusted the text in two places to clarify this:

'To test this hypothesis we inactivated *comM*, which promotes a division delay in competent cells¹⁵, as well as *cbpD*, the gene encoding the fratricide hydrolase CbpD, to which ComM provides immunity¹⁴. We investigated the antibiotic panel with this *comM* *cbpD*⁻ mutant and the isogenic parent strain *comM*⁻ *cbpD*⁺.'

'A *comM* mutant showed similar tolerance ratios to a *comM* *cbpD* double mutant, while a *cbpD* mutant showed similar tolerance ratios to wildtype, (Figure S13CDE). This showed that fratricide was not detected in these conditions, and demonstrated that CbpD plays no role in tolerance, with ComM alone responsible for this phenotype.'

8. Lines 336-7 state that propagation leads to exponential responses while AI diffusion leads to a linear response. Wouldn't the inherent positive feedback of CSP production also lead to an exponential response for the diffusion model? Would 'chain reaction' be a helpful way to describe the phenomenon?

We agree with the reviewer that at the individual cell level, the ComABCDE positive feedback loop would dramatically increase CSP production. This will occur only if a threshold of CSP is reached, a condition linked, among other things, to the affinity between CSP and ComD and the concentration of free CSP. With the classical QS model, the free diffusion of the AI leads to its accumulation in the medium until a threshold concentration is reached. The time required for CSP to accumulate and reach this threshold is gradual, fitting a linear equation. Of course, once the threshold is reached, due to the positive feedback loop, the rate of competence induction (X_B) should be the same whatever the cell density, since each cell should react with synchronicity. We have added a model illustrating classical QS to Figure S1 for comparison and better comprehension. Comparing Figures 1 and S1 allow better appreciation that for classical QS, the time until induction (X_A) is variable and dependent on cell density, while the rate of competence development (X_B) is constant. In contrast, for SI&P, the X_A time is independent of cell density, while the X_B rate depends on the density.

Chain reaction is an interesting term to use, but we feel it is more linked to physics, such as a nuclear chain reaction, and also may be misinterpreted as linear (like a chain), whereas the term propagation is more akin to the spread of a contagious virus through a population by contact between individuals, and directly links to an exponential equation.

To clarify the reference to 'linear' in the text, we have elaborated so the sentence now reads:

'Propagation becomes exponential if the AI is distributed from one SI cell to several P cells (Figure 1). This is in contrast with classical QS AI diffusion, where passive accumulation of AI leads to a linear response where all cells respond in synchrony once a threshold concentration of free AI is reached (Figure S1).'

9. Is anything in the literature known about MMR induction during competence development? Are any of the MMR genes within the ComE or ComX regulon?

Unexpectedly, we found that MMR is hyper-active in tolerant cells. However, highly sensitive RNAseq studies have shown that neither MMR genes *hexA* or *hexB* are induced during competence (Slager et al., 2019), corroborating earlier and less sensitive studies characterizing the transcriptional

competence regulon (Peterson et al., 2004, Dagkessamanskaia et al., 2004). We have added this important information to the revised version of the manuscript.

Minor

1. Line 186, the term 'artificially self-induced' doesn't seem appropriate. Aren't cells either artificially induced OR self-induced?

We agree, and have reworded in light of our further definition of SI and P cells previously in the text to read:

'Artificial SI cells'

2. Line 215, would help to indicate in the text the strain names that cannot naturally induce competence. Tables were not included in the supplemental information. Not a major issue for the review but be sure to include in final versions.

We apologise for the missing Tables, these have been added to the re-submission. We have also added a reference to the Materials and methods section as well as Table 3 for these strains.

3. Line 271, "...in other bacteria could play..." seems like the word 'and' is needed.

This is correct, we have reworded to read:

Increased tolerance has been associated with reduced growth and altered metabolism in other bacteria⁴⁸⁻⁵², **which** could play a role in competence-mediated tolerance increase.

4. Line 273, instead of 'screen' and 'isogenic wildtype', perhaps '...repeated the antibiotic panel with the isogenic strain.'

We agree, and have reworded this sentence.

5. Just want to be sure strain R4950 construction is described in the final version.

This is a typo which has been corrected. The strain should read R4590, which is already described.

We thank the reviewer for their vigilance.

6. Wondering why some graph lines in figure S2B (and others) are not connected.

It was not possible to connect these lines because at such low levels, stochastic heterogeneity means that some RLU values are negative after being corrected by the negative control, and can thus not be used to calculate a positive RLU/OD value. These values were removed from the data. We have clarified this in the Materials and Methods as follows:

'In some cases during the X_A period, the RLU values are below the level of sensitivity of the luminometer and provide negative values when blank values are subtracted. We were unable to calculate RLU/OD values from these values, which were removed from the analyses, resulting in occasional gaps between data points during the X_A period.'

Reviewer #2 (Remarks to the Author):

In this work, the author revisits the mechanism of competence induction and role of competence in pneumococci. Through a series of experiments, the authors conclude that competence is a

“population health sensor” in pneumococci, in which self-activation of competence in a small fraction of cells in the population in turn leads to propagation of competence throughout a population via cell-cell contacts - in a stress-related manner. It is suggested that this a mechanism to promote heterogeneity and diversity in the population, providing protection towards different stressors, as the competent state is also shown to alter the sensitivity of the population to antimicrobial agents. In line with previous reports, competence increase the tolerance to some antimicrobials, and interestingly it is here shown that this tolerance is modulated by the timing of stress induction and competence induction - and it is shown that the ComM protein is essential for this protection. The results also show that transformation efficiencies and properties are altered in such tolerant cells.

The manuscript suggests a new way to view the function of pneumococcal competence. The data overall seem robust and the paper includes a large number of thorough and rather complicated experiments. Although I find the results and models proposed interesting, some of the statements and conclusions should be moderated (see below). In addition, several points needs clarification.

We thank the reviewer for their stimulating comments regarding our work, and for their pertinent proposals that have undoubtedly helped us to improve our article. We have answered all concerns raised point by point in blue below.

Main points:

1. L138-9 and Supplementary information. The manuscript starts with an investigation of the differences in results and conclusions between the authors’ previous work (concluding that competence is propagated through cell-cell contact) and another study (Moreno-Gamez et al., concluding that competence is activate in the population at a certain AI concentration without the need a cell-cell contact). A number of experiments comparing the media, strains and luminometers are performed, and the authors conclude the results differ due to differences in reporter constructs and sensitivity of luminometers.

a. L939 – 942. Were the two D39-strains genotypically identical (except for the different reporter constructs)? It is not clear to me whether the two reporter constructs were compared in the exact same strain background (I cannot find any strain list or methodology describing this).

We apologise for the missing tables in this submission, which have now been added. In Table S3, the identity of these strains (D39V from Lausanne, D39 from Toulouse) is noted. We cannot confirm that there are no differences since our strain is not sequenced, but we note identical competence development profiles in the two strains using the same reporter construction (as described in Fig S2 & Table S3).

b. Fig. S1. Was the growth (OD) of the strains monitored during this experiment? Please include information about the growth.

We included OD curves in Figure S3A when differences in growth were observed between different media. In all other cases, no significant differences were observed, so we did not include these OD curves. However, we have now included growth curves in Figure S2A as requested.

c. Explain precisely how Xb and Xa was calculated in the Methods-section or first time it appears in the figure.

We have added the following explanation in the M&M section.

'The X_A and X_B values were calculated as previously described in Prudhomme 2016. Briefly, the theoretical regression of each slope before and during the first wave of propagation was calculated. The intersection of these slopes defines the X_A value (in minutes), and the rate of the exponential slope during propagation defines the X_B .'

d. Comparison of luminometers. What is meant by "reference luminometer"?

The reference luminometer is an Anthos Lucyl luminometer which we historically used for luminometry, chosen among 9 luminometers tested as having the best performance for our purpose of investigation. We refer to this as a 'reference' luminometer, as when we replaced it with a newer luminometer, we tested several in comparison to it, but not directly to each other. It is from these tests that the data in Figure S3 were taken, which explains why we were unable to compare the VarioSkan and Tekan luminometers directly. We have added the name of the reference luminometer to the text, along with the bibliographic reference, but we feel that an explanation of this in the text would complicate and is not necessary.

e. Line 951: please elaborate on how these results can "confirm that propagation is the general mechanism".

Differences in genotype and hardware led Moreno and colleagues to propose a different mode of competence development to our own. In this paragraph, we demonstrate why these conclusions were reached, and how they are inaccurate. This disproves their model, whilst fitting with our proposed model, which at this stage of our study remained to be validated by further experiments. However, our interest is not to emphasize the discreditation of their work but to confirm ours, hence why we phrased the text as such. When we talk of general mechanism, this is to highlight that there are not two mechanisms dependent on conditions/strains. We agree that this can be confusing, and so we have rephrased as such:

'Altogether, these results show that competence development does not fit a classical QS model, but still support a model of propagation among the population as the general mechanism of competence development in pneumococci, as previously proposed³⁹.'

f. What is not addressed clearly here is the cell-cell contact hypothesis. In Moreno-Gamez et al. it is shown (using microscopy) that competence could spread between cell/colonies without the need of cell-cell contact. Have such an experiment been tested here? How can it be excluded that competence propagation without cell-cell contact could play a role under certain conditions as well?

Timing is crucial to observe propagation by cell-to-cell contact in the first wave of competence. Without doubt, Moreno-Gamez et al show competence activation of a cell islet that must receive exogenous CSP from competent cells of a distant islet without direct contact (Movies S2 and S5). However, the receiver cell islet develops competence long after the competence shutoff of the donor. This indicates that after a first wave of competence, some CSP can be released and reach the receiver cell islet in the agar pad. Our experiments are different and not in contradiction with this observation since we focus exclusively on the very first wave of competence induction, and not beyond. In our point of view exploring competence after the first wave of induction would need to be done with great caution. Indeed, as we show in this article, several states of competence coexist

in the cell population, and the proportions of these heterogeneous cell states remain poorly defined. This would render any interpretations on how competence develops in subsequent waves difficult. Furthermore, our experiments are carried out on planktonic cells, while the movies reported in the Moreno-Gamez publication are carried out on agar pads, where the properties of CSP diffusion are not fully investigated. Thus, we interpret this experiment as pointing to a post-competence CSP release which is not antagonistic to our model of initial competence development.

We have added the following to the supplementary text:

‘In addition, Moreno-Gamez and colleagues reported the transmission of CSP between two separate cell islets on an agarose pad, one of which was unable to produce CSP³⁸. This indicated competence activation of cell islets receiving exogenous CSP from producing cells without direct contact, but by diffusion. However, the timing of this CSP transmission shows a delay greater than a competence cycle (30-100 min) between induction of the CSP producing cells and the non-producers. These experiments show that after a first competence wave, some CSP can be released and reach the receiver cell islet in agar pads. Our planktonic experiments are not in contradiction with this observation since we focus exclusively on the very first wave of competence induction, which we showed previously to rely on cell-to-cell contact.’

2. The term “population health sensor”

a. Throughout the paper it is stated that the results “revealed” competence “population health sensor”. I think this is an interesting model based on the data shown here, but some of these statements should be moderated. It is not yet known whether such a mechanism will work under more in-vivo like conditions. I also find “health” sensor is also too broad, as competence only will develop as response to some stressors. And isn’t it rather a stress sensor than a health sensor?

We chose populational health sensor over stress sensor, which for us does not refer to quite the same notion. However, we agree that how ‘health sensor was defined lacked clarity. Populational health sensor does not simply refer to the fact that certain stresses induce competence, but rather to how the SI&P mechanism allows a population of pneumococcal cells to differentiate between false alarm (when the levels of SI cells remains below a certain threshold) and actual stress (when the fraction of SI cells passes this threshold and promotes induction of competence by propagation). This refers more to the mode of competence induction defined in this study than to competence as a general stress response. In our opinion, this definition provides more nuance and fits better with the new data we bring. As a result we would like to maintain the notion of populational health sensor in the manuscript. We have altered the text at the first mention of health sensor in the results section to clarify the meaning of populational health sensor. The text now reads:

‘We unveil competence as a sensitive mechanism allowing individual cells to self-induce (SI) in response to stress, but only propagate (P) the signal in the face of sufficient stress. We call this mechanism Self-Induction and Propagation (SI&P). We have revealed that a growing pneumococcal culture is bimodal, with a minority of cells self-inducing competence in response to stress (SI cells) and a majority of cells not reaching a threshold of stress high enough to stimulate self-induction and thus remaining non-competent, but able to be induced by propagation (P cells). This mechanism allows a pneumococcal population to gauge the threat of a stress, where the proportion of SI cells in a population dictates whether propagation of competence throughout a population occurs or not. This allows discrimination between genuine stress threat and false alarm. SI&P thus shapes pneumococcal competence as a population health sensor, able to evaluate the health of a population via environmental stress levels and react rapidly at the populational level only once a certain level of stress is reached.’

b. Discuss/explain how stress mechanistically will affect the SI & P model proposed. Will it affect both the self-induction rate and the propagation, or both? Will all the stressors work through modification of the ComE-P pool? How does this proposed mechanism relate to the previously published mechanism for competence regulation by antibiotics (eg. PMID: 21933920, PMID: 32130952, PMID: 24725406).

Stress is individually sensed in each cell and an increase in stress results in an increase in self-inducing competent cells in the growing population. This affects the X_A time, as the threshold of self-inducing cells will be reached more rapidly, leading to earlier propagation. This can be seen for streptomycin. In contrast, MMC could impact both the X_A and X_B values, with X_A altered in the same way as streptomycin, but X_B potentially altered too since MMC kills cells, and a higher cell density results in faster propagation. However, we did not add this discussion to the text as it does not provide clear answers and could impact the flow of the paper.

Indeed, previous studies have identified multiple inducing cues and mechanisms regulating competence at the single-cell level (ori-ter ratio, replication fork arrests, ribosomal error rate, PMF inhibition). In this manuscript, we provide insight into how competence is coordinated at the level of the cell population, which is found to occur irrespective of the stress that generates the initial self-inducing cell fraction. We feel that in depth discussion of this point would not be relevant to the main messages of this study.

In addition, all studies conducted so far on the pneumococcal competence regulation mechanism point to the ComE/ComE~P ratio within a cell as being the main contributing factor controlling competence induction, firstly as to whether a cell self-induces, and also as to whether non-self-inducing cells are susceptible to the propagation signal. We have elaborated on this in the discussion, as detailed above.

3. Fig. 2B (upper panel). No error bars are shown. Include a measure on the variability between experiment related to the detection of this transformable subpopulation. There also appear to be some timepoints where no transformants were detected. Were this the same in all replicate experiments?

This experiment is a single experiment representative of triplicate repeats (as mentioned in the legend). Indeed, there is variability in the timing and detection of transformants that would complicate interpretation of the data around this point if error bars were included. In fact, the non-stressed cultures in Figure 2B and Figure S9 are identical replicates, showing differences in detection of SI cells. This is because these cells are detected just above the limit of detection of this assay. As a result, we propose to keep this representation to avoid confusion around this important point.

4. Line 200. How does these results “further validate” the mechanism proposed? Please elaborate.

We have restructured this paragraph for clarity, and have moderated our conclusion as requested. It now reads:

‘This fraction can promote propagation of the competence signal through a population, **further supporting** SI&P as the mode of regulation of the ComABCDE QS system.’

5. Fig. 3A. "Tolerance ratios were calculated by comparing cfu of stressed and non-stressed populations." Please be more specific in the definition of tolerance ratios.

We agree that this could have been worded in a clearer manner. To clarify, we have altered the sentence as follows:

'Tolerance ratios were calculated by first determining the survival percentage of stressed and non-stressed populations of competent and non-competent cells via colony counts, and then dividing the survival percentage of competent cells by that of non-competent cells.'

6. Tables S1 – S3 are lacking.

We apologise for this oversight, which has been rectified.

7. Fig. 3 and Fig. 4. Specify how the drug concentrations (relative to MIC) was standardized for the drugs.

The concentrations used were indicated in Table S1, which was missing. We have added this Table and added details on how the MIC were measured and the concentrations used in the survival assays chosen to the M&M section.

8. L348. The results show that competence increases tolerance to most drugs, except aminoglycosides streptomycin and kanamycin, but there is no further explanation about this. Can the authors (in the discussion section) speculate about potential reasons for this difference? (both in terms of mechanism and if there is any potential ecological relevance to this).

We have indeed thought about this, and have a proposal for the mechanism. Internalisation of aminoglycosides is increased when the proton motive force (PMF) is high (Webster and Shepherd 2022). In addition, the PMF has been shown to be increased specifically during pneumococcal competence (Lopez et al, 1989). Thus, we suggest that this increases the cytoplasmic concentration of aminoglycosides, rendering competent cells more sensitive. We have added this proposal to the discussion. However, without knowing the exact mechanism underlying this exacerbated sensitivity of competent cells to aminoglycosides, we have no clue regarding it's ecological relevance.

9. Fig. 4BC, Fig. S12 and related text related to the comM mutants

a. Complementation assays (eg. using inducible comM expression) should be performed for at least some of the antibiotics to confirm that the phenotypes can be complemented.

We have carried out complementation assays using inducible *comM* expression for two unlinked antibiotics (Ampicillin and Tetracycline). In both cases, the ectopic induction of *comM* in competent *comM* cells complemented the absence of *comM*. This confirms that the observed phenotype is principally associated with ComM.

We have added this data into the paper as Figure S14.

We have altered the text as follows:

'Complementation of the absence of *comM* by the ectopic *CEP_R-comM* construct^{18,53} restored the increased tolerance of competent cells to ampicillin or tetracycline, confirming that ComM is a main actor of competence-induced tolerance to stress (Figure S14).'

b. L356. It has previously been shown that a *comM* mutants have a cell division defect (Berge 2017). Here, the authors show that drug tolerance is reduced in the absence of *comM*. Can the author really conclude that the effect of *comM* related to antibiotic tolerance is “specific”, or could it be because the cells are already “weakened” in the absence of *comM*? Previous literature on the function of ComM should be discussed in the light of the new results obtained here.

The absence of ComM does not engender a cell division defect. In fact, induction of *comM* expression during competence mediates a delay in cell division, resulting in a programmed pause in the cell cycle (Bergé 2017). Cells lacking *comM* do not show any growth defect and do not appear to be ‘weakened’; they no longer exhibit a programmed division delay during competence, which was found in this study to reduce their gained tolerance to many lethal drugs.

10. Do the authors think that the mechanism proposed here would occur also in other peptide-based AI two/three-component systems, or is this a mechanism specific to pneumococcal competence.

This is an interesting question, and we believe that this type of mechanism could apply to systems where the exported AI is retained on the surface of producing cells. This could be any AI that is in part hydrophobic, giving it a preference to stick to the membrane. In particular, an article by Gardan and colleagues (J. Bacteriol 2013) presents phenotypes of competence induction in different inoculum densities of *S. thermophilus* strikingly similar to our results. Further studies are required to confirm whether *S. thermophilus* competence induction fits our SI&P model. However, the inducing cues could differ between systems. We have added reference to *S. thermophilus* to the discussion:

‘However, *Streptococcus thermophilus*, which also regulates competence via the two-component regulatory system ComRS, was shown to develop competence in a manner which could fit the SI&P model proposed here ⁷⁰, although this remains to be investigated.’

Minor points.

L331. “health” instead of “heath”

Corrected.

L335-337. Provide reference or describe how AI diffusion leads to linear response.

With the classical QS model, the diffusion of the AI leads to accumulation until a threshold is reached. The time required for CSP to accumulate and reach this threshold is gradual, fitting a linear equation. Of course, once the threshold is reached, due to the positive feedback loop, the rate of competence induction (X_B) should be the same whatever the cell density, since each cell should react with synchronicity. We have added a model illustrating classical QS to Figure S1 for comparison and better comprehension. Comparing Figures 1 and S1 allow better appreciation that for classical QS, the time until induction (X_A) is variable and dependent on cell density, while the rate of competence development (X_B) is constant. In contrast, for SI&P, the X_A time is independent on cell density, while the X_B rate depends on the density.

To clarify the reference to ‘linear’ in the text, we have elaborated so the sentence now reads:

'Propagation is exponential since the AI is retained on the cell surface, unlike AI diffusion, which leads to a linear response where all cells respond in synchrony once a threshold concentration of free AI is reached.'

L350. What does "...” mean?

Here we meant 'and other cellular processes', but we decided to remove this in any case to avoid confusion or lengthening the text unnecessarily.

P13. The Engelmoer&Rozen paper (doi: 10.1111/j.1558-5646.2011.01402.x) should also be cited in the discussion.

This citation has been added to the start of the discussion.

Fig. 5. Explain the term "Self-induction in response to breakdown" and "Next generation scattering".

We agree with the reviewer that these terms should be altered for clarity.

The first term was altered for clarity to read 'Self-induction in response to individual stress'. This represents for example metabolic breakdown at the individual cell level, and falls under the bracket of what we call false alarm, which we have added as a term to this step and explained in the Figure 5F legend.

The second term, as well as the other two under the bracket of multi-level heterogeneity, were altered to fit better with the definition of multilevel heterogeneity used in the discussion. They now read 'physiological state', 'genetic heredity', and 'physiological heredity'. This should increase clarity and streamline the figure with when this notion is referenced in the text. The legend of Figure 5F was also adjusted in light of this.

L394 and L399. The same claim on pneumococcal competence as a health sensor is repeated twice within 5 lines.

The first of these has been removed for clarity.

Line 191-192 and Fig. 2D. It would be useful to see examples of plotted data from this experiments used to calculate the X_A .

We have added this data as Figure S9C. The X_A was defined as described in the M&M section.

REVIEWERS' COMMENTS

Reviewer #1 (Remarks to the Author):

I am very satisfied with the authors' responses to my concerns. Several adjustments have been updated in the text and additional figures provided. I have no further concerns.

Reviewer #2 (Remarks to the Author):

The questions raised in the original review have been clarified and answered satisfactorily.